# Technical note: Evaporating water is different from bulk soil water in $\delta^2$H and $\delta^{18}$O and implication for evaporation calculation

Hongxiu Wang[1,2,3], Jingjing Jin[2], Buli Cui[1], Bingcheng Si[1,3], Xiaojun Ma[4], Mingyi Wen[2]

[1] College of Resources and Environmental Engineering, Ludong University, Yantai, Shandong Province 264025, China

[2] Key Laboratory of Agricultural Soil and Water Engineering in Arid and Semiarid Areas, Ministry of Education, Northwest A&F University, Yangling, Shaanxi Province 712100, China

[3] Department of Soil Science, University of Saskatchewan, Saskatoon SK S7N 5A8, Canada

[4] Gansu Provincial Department of Water Resources, Lanzhou, Gansu Province 730000, China

*Correspondence to*: Bingcheng Si (bing.si@usask.ca) and Buli Cui (cuibuli@163.com)

**Abstract.** Soil evaporation is a key process in the water cycle and can be conveniently quantified using $\delta^2$H and $\delta^{18}$O in bulk surface soil water (BW). However, recent research shows that soil water in larger pores evaporates first and differs from water in smaller pores in $\delta^2$H and $\delta^{18}$O, which disqualifies the quantification of evaporation from BW $\delta^2$H and $\delta^{18}$O. We hypothesized that BW had different isotopic compositions from evaporating water (EW). Therefore, our objectives were to test this hypothesis first and then evaluate whether the isotopic difference alters the calculated evaporative water loss. We measured the isotopic composition of soil water during two continuous evaporation periods in a summer maize field. Period I had a duration of 32 days following a natural precipitation event, and Period II lasted 24 days following an irrigation event with a $^2$H-enriched water. BW was obtained by cryogenically extracting water from samples of 0–5-cm soil taken every 3 days; EW was derived from condensation water collected every 2 days on a plastic film placed on the soil surface. The results showed that when event water was "heavier" than pre-event BW, $\delta^2$H of BW in Period II decreased with an increase in evaporation time, indicating heavy water evaporation. When event water was "lighter" than the pre-event BW, $\delta^2$H and $\delta^{18}$O of BW in Period I and $\delta^{18}$O of BW in Period II increased with increasing evaporation time, suggesting light water evaporation. Moreover, relative to BW, EW had significantly smaller $\delta^2$H and $\delta^{18}$O in Period I and significantly smaller $\delta^{18}$O in Period II ($p < 0.05$). These observations suggest that the evaporating water was close to the event water, both of which differed from the bulk soil water. Furthermore, the event water might be in larger pores, from which evaporation takes precedence. The soil evaporative water losses derived from EW isotopes were compared with those from BW. With a

small isotopic difference between EW and BW, the evaporative water losses in the soil did not differ significantly ($p > 0.05$). Our results have important implications for quantifying evaporation processes using water stable isotopes. Future studies are needed to investigate how soil water isotopes partition differently between pores in soils with different pore size distributions and how this might affect soil evaporation estimation.

**1 Introduction**

Terrestrial ecosystems receive water from precipitation and subsequently release all or part of the water to the atmosphere through evapotranspiration. The evapotranspiration process consumes approximately 25% of the incoming solar energy (Trenberth et al., 2009) and can be divided into two components: transpiration from plant leaves and evaporation from the soil surface. Soil evaporation varies from 10 to 60% of the total precipitation (Good et al., 2015; Oki and Kanae, 2006). Precise estimation of soil evaporative water loss relative to precipitation is critical for improving our knowledge of water budgets, plant water use efficiency, global ecosystem productivity, allocation of increasingly scarce water resources, and calibrating hydrological and climatic models (Kool et al., 2014; Oki and Kanae, 2006; Or et al., 2013; Or and Lehmann, 2019; Wang et al., 2014).

Water loss from soil progresses with air invasion into the soil in the order of large to small pores (Aminzadeh and Or, 2014; Lehmann and Or, 2009; Or et al., 2013). Soil pores can be divided into large, medium, and small pores. There is a minimum amount of small pore water at which liquid water in soil is still continuous or connected, below which liquid water is hydraulically disconnected, and vapor transport is the only way to further reduce water in soil. This water content is called the residual water content in the soil characteristic curve (Van Genuchten, 1980; Zhang et al., 2015). When large soil pores are filled with water, water in small pores does not participate in evaporation (Or and Lehmann, 2019; Zhang et al., 2015). Therefore, soil evaporation can be divided into three stages (Hillel, 1998; Or et al, 2013). Stage I: the evaporation front is in the surface soil, and water in large and medium pores participates in evaporation, but larger pores are the primary contributors. With the progressive reduction of water in the larger pores, the evaporation rate gradually decreases. Stage II: evaporation front is still in the surface soil, but larger pores are filled with air, water residing in the medium soil pores in the surface soil evaporates, and deep larger soil pores recharge the surface medium pores by capillary pull

(Or and Lehmann, 2019), and the evaporation rate remains constant. Stage III: the hydraulic connectivity between the surface medium pores and deep large pores breaks, such that the evaporation front recedes into the subsurface soil. Water in the surface small pores and water in medium pores on the evaporation front evaporates. The evaporation rate decreases to a low value (Or et al, 2013).

Furthermore, water in small pores and large pores may differ in isotopic compositions. As is well-known,
pre-event soil water occupies the smallest pores. Depending on the rainfall amount and intensity, an event water may have three pathways. First, a subsequent small event water fills the empty small soil pores. Second, event water with small rates, but long duration, may also displace the pre-existing, saturated smaller pores with slow flow velocity (Beven and Germann, 1982; Brooks et al., 2010; Klaus et al., 2013; Sklash et al., 1996); in cases that the water flow into a relatively impermeable layer, the pre-event water
in smaller pores may be forced into large pores, due to the underlining hydraulic barriers (Si et al., 2017). Third, when the event water is large and intense, the event water preferentially enters large pores, bypassing the saturated small pores with large flow velocity (Beven and Germann, 1982; Booltink and Bouma, 1991; Kumar et al., 1997; Levy and Germann 1988; Radolinski et al., 2021; Sprenger and Allen, 2020). Because the exchange rate between these two flow domains is small (Šimůnek and van Genuchten
2008), small pores will lock the signature of first filling water. As the flow velocity is determined by the soil pore size, larger pores have greater hydraulic conductivity, and consequently water residing in larger pores flows faster and thus drains first. Conversely, water residing in small pores drains last (Gerke and Van Genuchten, 1993; Phillips, 2010; Van Genuchten, 1980). Therefore, soil water in smaller pores has a longer residence time or memory (Sprenger et al., 2019b), while water in large pores geneally have a
short memory. This differing memory between large pore and smaller pores, due to the sequence of water infiltration and drainage, could introduce variability in the isotopic composition between soil pore spaces. Additionally, due to seasonal, temperature, and amount effects of local precipitation events, there is strong temporal variation in the isotopic composition of precipitation (Kendall and McDonnell, 2012). As a result, precipitation events, differing in isotopic compositions, could recharge different soil pores,
which may yield isotopic heterogeneities in soil pore spaces (Brooks et al., 2010; Goldsmith et al., 2012; Good et al., 2015). Isotopically, small-pore water may be similar to old precipitation, with large-pore water resembling new precipitation (Sprenger et al., 2019a; Sprenger et al., 2019b).

The isotopic variations in the soil pore space could also result from mineral-water interaction, soil particle

surface adsorption, and soil tension (Gaj et al., 2017a; Gaj and McDonnell, 2019; Oerter et al., 2014; Orlowski and Breuer, 2020; Thielemann et al., 2019).

Despite the recent progress in understanding evaporation processes and isotope partitioning in soil pore space, the latter, to the best of our knowledge, is not considered in the calculation of soil evaporative water loss in terms of the isotope-based method. The isotopic composition of bulk soil water, which is extracted by cryogenic vacuum distillation, containing all pore water, is still routinely used in evaporation calculations using the Craig-Gordon model (Allison and Barnes, 1983; Dubbert et al., 2013; Good et al., 2014; Robertson and Gazis, 2006; Sprenger et al., 2017). This might bias the evaporation estimates because of isotopic variation in pore space and the preference for larger-pore water by evaporation. Therefore, we hypothesize that the isotopic composition in evaporating water (EW) is similar to that of water in larger pores but differs from that in BW; thus, evaporative water loss based on isotope values in BW will be biased. The objectives of this study were to verify 1) whether isotopic compositions differ between EW and BW and 2) if the isotopic composition difference substantially biases the calculated evaporative water loss. This study may help improve our understanding of soil evaporation and ecohydrological processes.

## 2 Materials and methods

### 2.1 Experimental site

The field experiment was conducted from June to September of 2016 at Huangjiabao Village (34°17′ N, 108°05′ E, 534 m above sea level), located in the southern Chinese Loess Plateau. The study site experiences a temperate, semi-humid climate, with a mean annual temperature of 13 °C, precipitation of 620 mm, and potential evaporation of 1,400 mm (Liang et al., 2012). Winter wheat followed by summer maize rotation is routine practice in this region (Chen et al., 2015).

### 2.2 Experimental design

A summer maize field (35 m long and 21 m wide) was selected for this study. On June 18, 2016, maize seeds were sown in alternating row spaces of 70 cm and 40 cm with 30-cm seed intervals in each row. Seeds were planted at a depth of 5 cm beneath the soil surface using a hole-sowing machine. On August 26, 2016, the field was irrigated with 30 mm water ($\delta^2H$ = 49.87 ± 2.7 ‰, $\delta^{18}O$ = -9.40 ± 0.05 ‰, $n$ = 5)

which was a mixture of tap water ($\delta^2H$ = -61.11 ‰, $\delta^{18}O$ = -9.42 ‰) and deuterium-enriched water (the $^2H$ concentration was 99.96%, $\delta^2H$ = 1.60 × $10^{10}$ ‰; Cambridge Isotope Laboratories, Inc., Tewksbury, MA, USA).

**2.3 Samples collection and measurement**

A randomized replication design was used to collect samples. To determine the water isotopic composition in EW from the condensation water of the evaporation vapor, we randomly selected three rectangular plots (40 cm long and 30 cm wide) in the field. A channel of 3 cm deep was dug around the edge of the plot (Fig. 1). Subsequently, a piece of plastic film without holes (approximately 0.2 $m^2$, 40 and 50 cm) was used to cover the soil surface, with an extra 5 cm on each side. The channels were then

backfilled with soil to keep the covered area free of the wind. To eliminate the secondary evaporation of the condensation water, we first allowed evaporation and condensation to equilibrate for 2 days under the plastic film. Then, in the early morning (approximately 7 a.m.), we collected the condensation water adhered to the underside of the plastic film using an injection syringe (Fig. 1a). The collected water was immediately transferred into a 1-mL glass vial. Therefore, it is reasonable to assume that the condensation

water was in constant equilibrium with the evaporating water in the soil, and the water isotopes of evaporating water in the soil could be obtained from condensation water on the plastic film. After collection, the plastic film was removed with little disturbance to the site. Subsequently, three new plots were selected randomly and similarly covered with a new piece of plastic film for the next water collection.

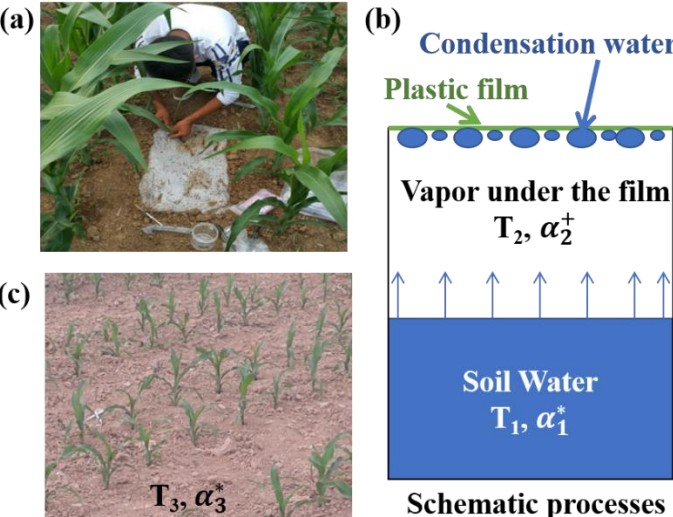

T$_1$: Soil temperature under film
T$_2$: Vapor temperature under film
T$_3$: Field soil temperature
$\alpha_1^*$: Equilibrium fractionation factor in soil under film
$\alpha_2^+$: Equilibrium fractionation factor in vapor under film
$\alpha_3^*$: Equilibrium fractionation factor in field soil

**Figure 1: Photo of new plastic film cover and condensation water collection using a syringe (a), schematic of the condensation process (b), and photo of field soil condition (c).**

In addition, BW was obtained from 0–5-cm surface soil water (Wen et al., 2016). The soil samples were collected using a soil auger every 3 days with 3 replicates, and each was mixed well and separated into 2 subsamples: one for determining the soil gravimetric water content and the other for water stable isotope analysis. The subsample for soil gravimetric water content was stored in an aluminum box and oven-dried for 24 h at 105 °C, while the water stable isotope analysis sample was stored in 150-mL high-density polyethylene bottles, sealed with Parafilm®, transported, and stored in a freezer at -20 °C at the laboratory until cryogenic liquid water extraction took place. To obtain bulk soil density, field capacity, and residual water content, three 70-cm deep pits were dug at the end of the growing season. Stainless rings with a volume of 100 cm$^3$ (DIK-1801; Daiki Rika Kogyo Co., Ltd, Saitama, Japan) were pushed into the face of each soil pit at depths of 10, 20, 40, and 60 cm to obtain the soil samples. The soil samples were then saturated with distilled water, weighed, and placed in a high-speed centrifuge (CR21GII; Hitachi, Tokyo, Japan) with a centrifugation rotation velocity equivalent to a soil suction of 1 kPa for 10 min. The soil samples were weighed again to obtain the gravimetric water content at the aforementioned suction. This was repeated for suctions of 5, 10, 30, 50, 70, 100, 300, 500, and 700 kPa for 17, 26, 42, 49, 53, 58, 73, 81, and 85 min, respectively, to obtain the soil characteristic curve. After centrifugation,

the soil samples were oven-dried and weighed to obtain the bulk soil density, which was used to convert gravimetric water content to volumetric water content.

A cryogenic vacuum distillation system (Li-2000; Lica United Technology Limited, Beijing, China) with a pressure of approximately 0.2 Pa and a heating temperature of 95 °C was used to extract soil water (Wang et al., 2020). The extraction time was at least 2 h until all the water evaporated from the soil and was deposited in the cryogenic tube. To calculate the extraction efficiency, samples were weighed before and after extraction and weighed again after oven-drying for 24 h following extraction. Samples with an

extraction efficiency of less than 98% were discarded. In terms of weight, cryogenic vacuum distillation extracts all water from the soil. However, in terms of isotopic compositions, the extracted water is generally depleted in heavy isotopes relative to the reference water, and the extent of depletion is affected by soil clay content and water content due to incomplete soil water extraction (Orlowski et al., 2016; Orlowski et al., 2013).To extract all water from a soil sample, a higher extraction temperature (>200 °C)

might be desirable, especially for soils with substantial clay particles such as in the present study (clay content of 0.24 g g$^{-1}$) (Gaj et al., 2017a; Gaj et al., 2017b; Orlowski et al., 2018). Therefore, the water isotopic compositions obtained from our distillation system were subsequently corrected by calibration equations:

$\delta^2H(post\ corrected)=\delta^2H(measured)\text{-}21.085*WC(water\ content)+5.144*CC(clay\ content)+5.944$ and

$\delta^{18}O(post\ corrected)=\delta^{18}O(measured)\text{-}2.095*WC+0.783*CC+0.502$. The equations were obtained through a spiking experiment with 205 °C-oven-dried soils.

    Five deep soil profiles were collected on July 17, 2016 (pre-precipitation), August 3, 2016 (10 days after precipitation, DAP), August 17, 2016 (24 DAP), September 1, 2016 (6 days after irrigation, 6 DAI), and September 16, 2016 (21 DAI) with increments of 0–5, 5–10, 10–20, 20–30, 30–40, and 40–60 cm. These

soil samples were used to measure soil texture (Dane and Topp, 2020), soil water content, and soil water isotopic composition. Furthermore, the lc-excess of the soil water before the enriched-$^2$H irrigation was calculated to infer the evaporation enrichment of soil water. A more negative lc-excess value indicates a stronger evaporation effect (Landwehr and Coplen, 2006).

$lc\text{-}excess= \delta^2H\text{-}7.81\delta^{18}O\text{-}10.42,$                                         (1)

where $\delta^2H$ and $\delta^{18}O$ are the soil water isotopic compositions; 7.81 and 10.42 are the slope and intercept of the local meteoric water line (LMWL), respectively.

Precipitation was collected during the entire growing season using three rainfall collectors (Wang et al., 2010) in the experimental field. The amount of rainfall was determined by weighing using a balance. Subsequently, sub samples of these rainfall samples were transferred to 15-mL glass vials, sealed immediately with Parafilm®, and placed in a refrigerator at 4 °C. To obtain the LMWL, we used 3 years of precipitation isotope data (Zhao et al., 2020) from April 1, 2015, to March 19, 2018. The equation for LMWL was $\delta^2H=7.81\ \delta^{18}O+10.42$.

Hourly air and 0–5-cm soil temperature under the newly covered plastic film from September 10, 2016, to September 28, 2016, were measured using an E-type thermocouple (Omega Engineering, Norwalk, CT, USA) controlled by a CR1000 datalogger (Campbell Scientific, Inc., Logan, UT, USA). The 0–5-cm field soil temperature was measured during the whole field season using an ibutton device (DS1921G; Maxim Integrated, San Jose, CA, USA) at a frequency of 1 h. The 0–5-cm soil temperature and air temperature under the plastic film are required to calculate the evaporation ratios, but these measurements were not available before September 10, 2016. To obtain these temperature values, a regression equation was established between the measured 0–5-cm soil temperature values under the newly covered plastic film and those without plastic film covering from September 10, 2016, to September 28, 2016. We then used the equation to estimate 0–5-cm soil temperature under the newly covered plastic film before September 10, 2016, based on the ibutton-measured temperature of the 0–5-cm soil without the plastic film covering in the same period. Subsequently, another regression equation was obtained between air temperature and 0–5-cm soil temperature from September 10, 2016, to September 28, 2016, both of which were under the newly covered plastic film. Then the air temperature under the newly covered plastic film before September 10, 2016, was estimated from the estimated 0–5-cm soil temperature under the newly covered plastic film. The regression equations are presented in the Supplement File. Moreover, the hourly ambient air relative humidity was recorded by an automatic weather station (HOBO event logger; Onset Computer Corporation, Bourne, MA, USA) located 3 km away.

A micro-lysimeter (Ding et al., 2013; Kool et al., 2014) replicated thrice, made of high-density polyethylene with a 10-cm in depth, 5.2-cm inner radius, and 3-mm thickness, was used to obtain the soil evaporation amount. The micro-lysimeter was pushed into the soil surface between maize rows to retrieve an undisturbed soil sample. Subsequently, we sealed the bottom, weighed the micro-lysimeter, placed it back in the soil at the same level as the soil surface, and no other sensor was installed in the micro-

lysimeter. After 2 days of evaporation, the lysimeter was weighed again. The mass difference was defined as the amount of soil evaporation. When evaporation occurs, unlike with soil outside the lysimeter, the soil within lysimeters is not replenished with water from deeper layers; thus, relative to soil outside the lysimeter, the soil water content within the lysimeters is generally smaller following continuous evaporation. Therefore, to represent the field soil conditions, the soil within the lysimeter was replaced every 4 days. In addition, after every rainfall or irrigation period, the inner soil was changed immediately. All water samples were analyzed for $\delta^2H$ and $\delta^{18}O$ using isotopic ratio infrared spectroscopy (Model IWA-45EP; Los Gatos Research, Inc., San Jose, CA, USA). The instrument's precision was 1.0 ‰ and 0.2 ‰ for $\delta^2H$ and $\delta^{18}O$, respectively. Three liquid standards (LGR3C, LGR4C, and LGR5C and their respective $\delta^2H$ = -97.30, -51.60, -9.20 ‰; $\delta^{18}O$ = -13.39, -7.94, -2.69 ‰) were used sequentially for each of the three samples to remove the drift effect. To eliminate the memory effect, each sample was analyzed using six injections, of which only the last four injections were used to calculate the average value. To check the effect of extrapolation beyond the range of standards, we performed a comparative experiment. In the experiment, 10 liquid samples with $\delta^2H$ varying from 0.14 to 107 ‰ and $\delta^{18}O$ from -1.75 to 12.24 ‰ were analyzed using LGR 3C, LGR 4C, and LGR 5C as standards (same with our former analysis) and were also analyzed using LGR 5C, GBW 04401 ($\delta^2H$ = -0.4 ‰, $\delta^{18}O$ = 0.32 ‰), and LGR E1 ($\delta^2H$ = 107 ‰, $\delta^{18}O$ = 12.24 ‰) as standards. The differences between the two sets of measurements were regressed with the sample isotope values obtained using LGR 5C, GBW 04401, and LGR E1 as standards, with a linear relationship of $\Delta^2H$ = -0.019$\delta^2H$-0.271 (with $R^2$=1) and $\Delta^{18}O$ = -0.053$\delta^{18}O$-0.091 (with $R^2$=1). We then applied the relationship and corrected the isotopic data that had $\delta^2H$ larger than -9.26 ‰ and $\delta^{18}O$ larger than -2.72 ‰. All the analyses in this study were based on the reanalyzed data.

The results are reported in $\delta$ notation:

$$\delta = \left( \frac{R_{sample}}{R_{standard}} - 1 \right) \times 1000 \text{ ‰} , \tag{2}$$

where $R_{sample}$ denotes the ratio of the number of heavy isotopes to that of the light isotope in the sample water, and $R_{standard}$ is the ratio in the Vienna Standard Mean Ocean Water (V-SMOW).

## 2.4 Equilibrium fractionation processes

The isotopic composition of EW was calculated using the condensation water that adhered to the underside of the newly covered plastic film. We assumed that the water vapor under the newly covered

plastic film and above the surface soil constitutes a closed system. Within the system, two equilibrium

fractionation processes are temperature-dependent and occur independently: evaporation from surface

soil water to air under the plastic film occurs during the day time (8 a.m. to 8 p.m., Fig. 2), condensation

from the water vapor under the plastic film to liquid water ensued at night time (8 p.m. to 8 a.m.), and

the resulting dews (condensation water) adhered to the plastic film. The average temperatures from 8 a.m.

to 8 p.m. and 8 p.m. to 8 a.m. on the day before water collection were used to calculate the equilibrium

fractionation factor ($\alpha$) (Horita and Wesolowski, 1994) for the evaporation and condensation processes,

respectively.

$$1000 \times \ln\alpha^+ \left(^2H\right) = \frac{1158.8 \times T^3}{10^9} - \frac{1620.1 \times T^2}{10^6} + \frac{794.84 \times T}{10^3} - 161.04 + \frac{2.9992 \times 10^9}{T^3} \ , \tag{3}$$

$$1000 \times \ln\alpha^+ \left(^{18}O\right) = -7.685 + \frac{6.7123 \times 10^3}{T} - \frac{1.6664 \times 10^6}{T^2} + \frac{0.35041 \times 10^9}{T^3} \ , \tag{4}$$

$$\alpha^+ = \frac{\delta_{liquid} + 1000}{\delta_{vapor} + 1000} \ , \tag{5}$$

$$\alpha^* = 1/\alpha^+ \ , \tag{6}$$

where $\alpha^+$ and $\alpha^*$ are the equilibrium fractionation factors during condensation and evaporation,

respectively; $\delta_{liquid}$ is the isotopic composition in the liquid water, $\delta_{vapor}$ is the isotopic composition in

the vapor, and $T$ is the temperature presented in Kelvins.

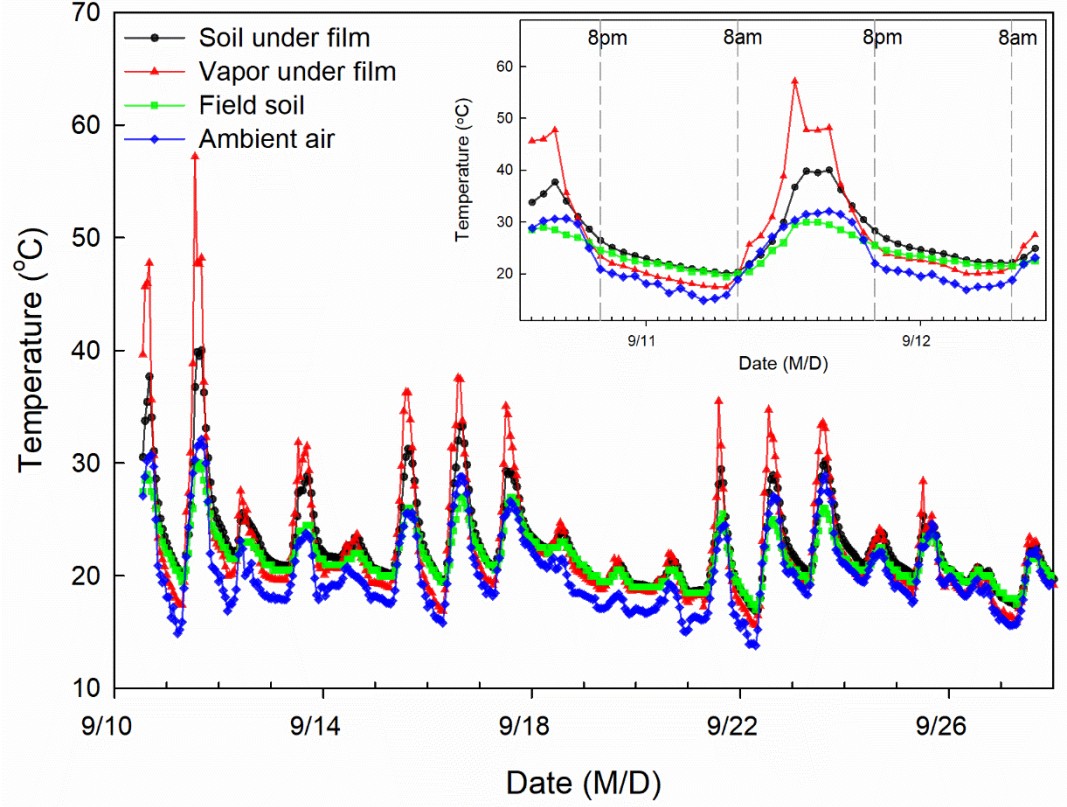

**Figure 2: Temporal variation in temperature of soil under film, vapor under film, field soil, and ambient air during the study period.**

Based on Eqs. (3) to (6) and Fig. 1b, the fractionation factors for the two processes under the newly covered plastic film are expressed using equations (7) and (8).

$$\alpha_1^* = \frac{\delta_{EW} + 1000}{\delta_{Vp} + 1000} \ , \tag{7}$$

$$\alpha_2^+ = \frac{\delta_{CW} + 1000}{\delta_{Vp} + 1000} \ , \tag{8}$$

where $\delta_{Vp}$ represents the isotope values of water vapor under the newly covered plastic film, $\delta_{EW}$ represents the isotope value in evaporating water, and $\delta_{CW}$ represents the isotope value in condensation water.

Combining equations (7) and (8), we obtain the isotopic composition in the EW:

$$\delta_{EW} = \frac{1}{\alpha_1^* \alpha_2^+} (\delta_{CW} + 1000) - 1000 \ , \tag{9}$$

### 2.5 Evaporative water losses

For an open system (field soil condition, Fig. 1c), evaporation from surface soil water to ambient air undergoes two processes: the equilibrium fractionation process from the surface soil to the saturated

vapor layer above the soil surface and the kinetic fractionation process from the saturated vapor layer to

ambient air. The isotopic composition of evaporation vapor is controlled by the isotope values of the

evaporating soil water and ambient vapor, equilibrium, and kinetic fractionations. The kinetic

fractionation can be described by the enrichment factors ($\varepsilon_k$) of $^{18}$O and $^2$H as a function of ambient air

relative humidity ($h$) (Gat 1996):

$$\varepsilon_k(^{18}O) = 28.5(1 - h), \tag{10}$$

$$\varepsilon_k(^2H) = 25.115(1 - h), \tag{11}$$

The total enrichment factor, $\varepsilon$, can be obtained from the kinetic enrichment factor ($\varepsilon_k$) and equilibrium

fractionation factor ($\alpha_3^*$) (Skrzypek et al., 2015):

$$\varepsilon = (1 - \alpha_3^*) * 1000 + \varepsilon_k, \tag{12}$$

The ambient vapor isotopic composition ($\delta_A$) can be obtained as follows (Gibson et al., 2008):

$$\delta_A = (\delta_{rain} - (\alpha_A^+ - 1) * 1000)/\alpha_A^+ , \tag{13}$$

where $\alpha_A^+$ is the equilibrium fractionation factor in the ambient air, $\delta_{rain}$ is the amount weighted

isotopic composition in precipitation from July 11, to September 16, 2016.

The isotopic compositions of bulk soil water and evaporating water can be used to evaporating soil water

in the Craig-Gordon model (Eq. 14) to calculate the isotope value of the evaporation vapor ($\delta_{EV}$).

$$\delta_{EV} = \frac{\alpha_3^* \delta_{BW} - h\delta_A - \varepsilon}{(1-h) + \varepsilon_k/1000} \text{ or } \frac{\alpha_3^* \delta_{EW} - h\delta_A - \varepsilon}{(1-h) + \varepsilon_k/1000} \tag{14}$$

Based on the bulk soil water isotope mass balance, i.e., the change in bulk soil water isotopic composition

multiplied by the soil water reduction equals the evaporation vapor isotopic composition multiplied by

the evaporation amount (Hamilton et al., 2005; Skrzypek et al., 2015; Sprenger et al., 2017), we can

calculate evaporative water loss to the total water source ($f$).

$$f = 1 - \left[\frac{\delta_{BW} - \delta^*}{\delta_I - \delta^*}\right]^{\frac{1}{m}} , \tag{15}$$

where $\delta_I$ is the isotopic signal of the original water source. $\delta_I$ is generally unknown and can be

conveniently obtained by calculating the intersection between the regression line of the 0–5-cm bulk soil

water isotope in Period I and the LMWL in the dual-isotope plot (Fig. 3). $m$ and $\delta^*$ in Eq. (15) are

given by:

$$m = \frac{h - \frac{\varepsilon}{1000}}{1 - h + \frac{\varepsilon_k}{1000}} , \tag{16}$$

$$\delta^* = \frac{h * \delta_A + \varepsilon}{h - \frac{\varepsilon}{1000}} , \tag{17}$$

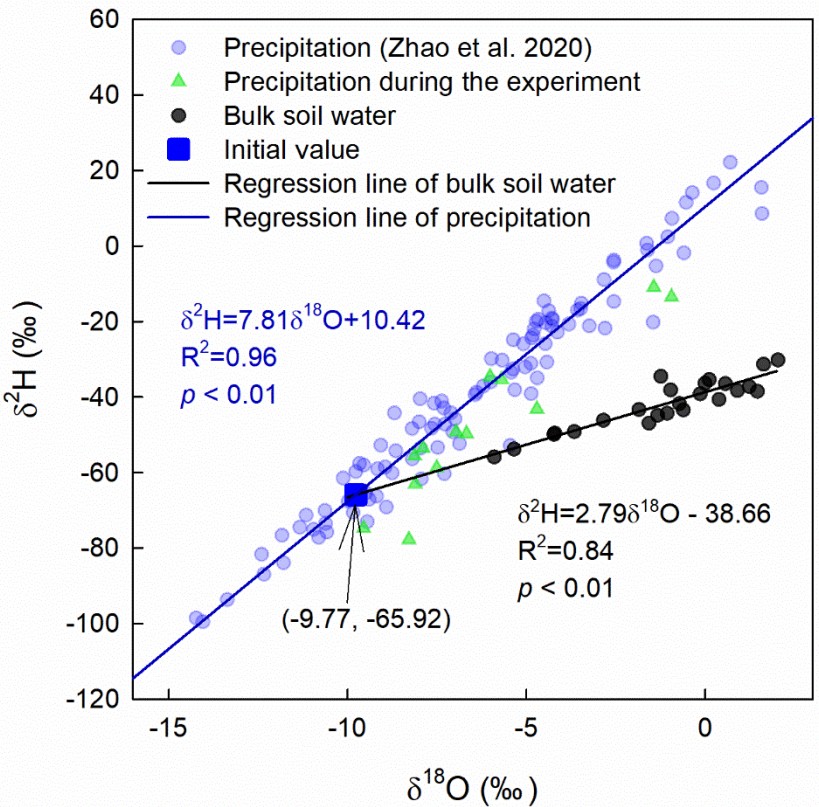

**Figure 3: The dual-isotope plot of precipitation and 0–5-cm bulk soil water from 25 July, 2016, to 25 August,**

**2016 (Period I). The regression line of precipitation represents the local meteoric water line.**

In Period II, the initial values (-9.52 and 11.50 ‰ for $\delta^{18}O$ and $\delta^2H$, respectively) were calculated from

the weighted average of the isotope values of irrigation water and Period I original water described above.

To calculate evaporative water loss from EW $\delta^{18}O$, we used BW to express EW and obtained the

following formulas (Eqs. 18–19) for evaporative water loss.

$f = 1 - \left[ \frac{\delta_{BW} - \delta^* + n}{\delta_I - \delta^* + n} \right]^{\frac{1}{m}}$ ,                                                                (18)

where $n$ is an intermediate variable and can be expressed as follows:

$n = \frac{-1.99\alpha_1^*}{h - \frac{\varepsilon}{1000}}$ ,                                                                                           (19)

**2.6 Statistical Analysis**

A general linear model (GLM) was used to test if the regression lines for isotopic

composition/evaporative water loss of BW as a function of days after precipitation/irrigation (DAP/I)

differ from those of EW. GLM was also used to compare the Period I evaporative water loss derived from

$\delta^2H$ and $\delta^{18}O$ of BW. The Shapiro-Wilk test was used to test the normality of the error structure of the

model ($p > 0.05$). Further, Student's $t$-test (Knezevic, 2008) was used to compare two corresponding

mean values of three replicates.

## 3 Results

### 3.1 Variation of 0–5-cm soil water content

Between the two large precipitation events on July 24, 2016, and September 20, 2016, there was no effective precipitation, except for an irrigation event of 30 mm on August 26, 2016 (Fig. 4a). Thus, two continuous evaporation periods can be identified: Period I from July 25, 2016, to August 25, 2016, and Period II from August 27, 2016, to September 19, 2016.

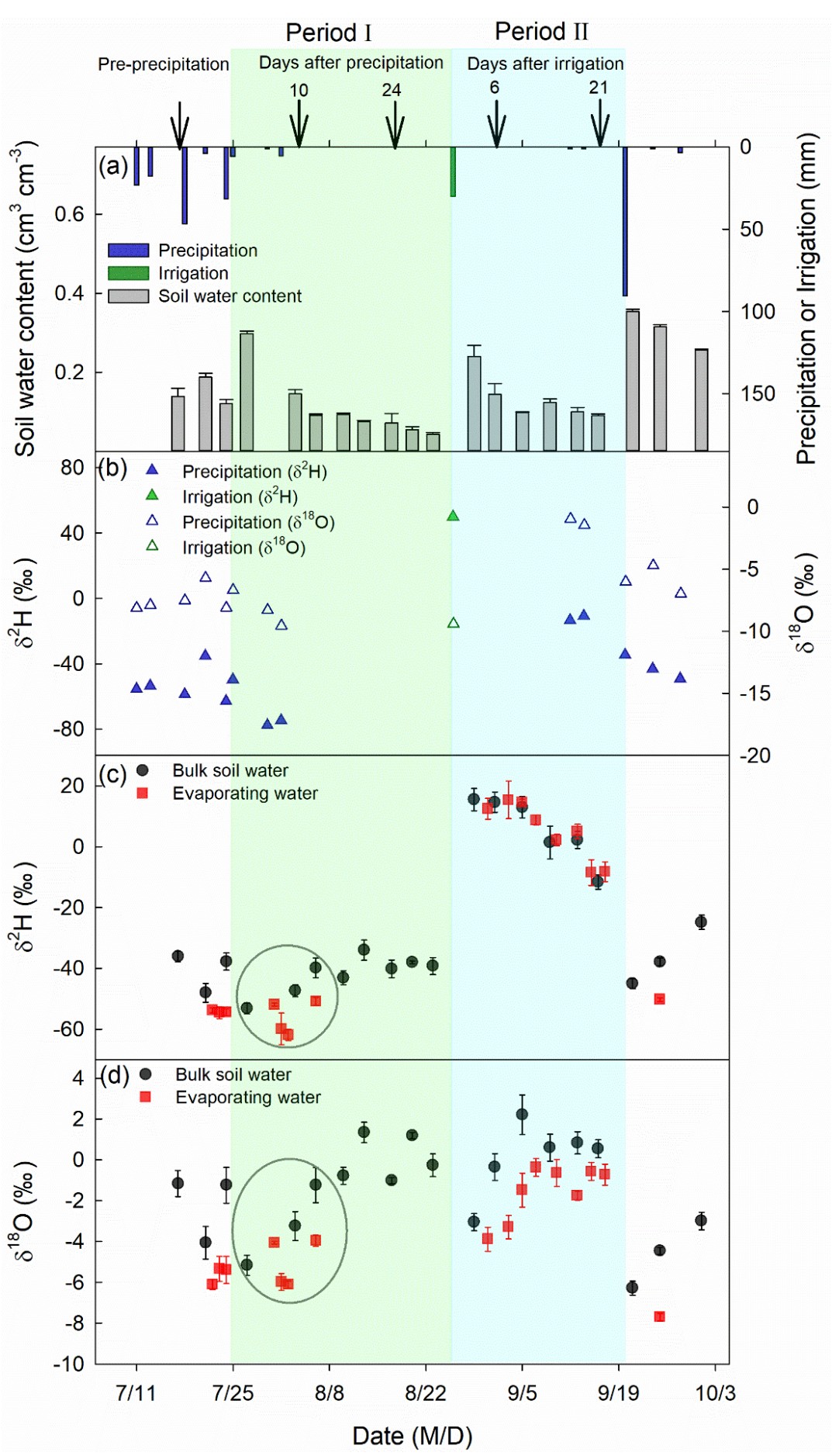

**Figure 4: The amount of precipitation, irrigation, and 0–5-cm bulk soil water content (a), $\delta^2H$ and $\delta^{18}O$ of precipitation and irrigation (b), $\delta^2H$ of 0–5-cm bulk soil water and evaporating water (c), $\delta^{18}O$ of 0–5-cm bulk soil water and evaporating water (d) at different times of the experimental period. Black arrows in panel (a) indicate dates when deep soil sampling took place, and the corresponding days after precipitation (irrigation) are indicated above the arrows. The two evaporation periods, marked by colored shades, include Period I from July 25, 2016, to August 25, 2016 (green) and Period II from August 27, 2016, to September 19, 2016 (cyan). Within the green circle in Period I, the mean ± standard error values were $\delta^2H$ =-46.80 ± 1.07 ‰ and $\delta^{18}O$ -3.22 ± 0.31 ‰ for 0–5-cm bulk soil water, and $\delta^2H$ =-57.55 ± 2.60 ‰ and $\delta^{18}O$ = -5.35 ± 0.22 ‰ for evaporating water.**

Soil water content in 0–5 cm reached field capacity (0.30 $cm^3$ $cm^{-3}$) with a volumetric water content of 0.30 ± 0.007 $cm^3$ $cm^{-3}$ and a porosity of 0.50 ± 0.05 $cm^3$ $cm^{-3}$ right after the first large precipitation event (July 24, 2016) and then decreased with evaporation time (grey bars in Fig. 4a). At the end of Period I, 0–5-cm soil water content was 0.05 ± 0.005 $cm^3$ $cm^{-3}$, close to the residual water content of 0.08 ± 0.03 $cm^3$ $cm^{-3}$. Similarly, after the irrigation event (August 26, 2016), 0–5-cm soil water content increased to a high value (0.24 ± 0.03 $cm^3$ $cm^{-3}$) and then decreased with an increase in evaporation time (Fig. 4a). At the end of Period II, 0–5-cm soil water content was 0.09 ± 0.005 $cm^3$ $cm^{-3}$, also close to the residual water content. In total, there was a 12.73 ± 0.58 mm and 7.51 ± 1.24 mm reduction in soil water storage at 0–5 cm during Periods I and II, respectively. However, from the micro-lysimeters, we obtained a total evaporation amount of 20.45 ± 0.95 mm in Period I and 9.56 ± 1.18 mm in Period II. Therefore, the evaporation amount in each of the two periods was greater than the soil water storage reduction at 0–5 cm, suggesting that soil water from below 5 cm moved up and participated in evaporation in each of the two periods, especially in Period I.

### 3.2 $\delta^2H$ and $\delta^{18}O$ in evaporating water and bulk soil water

The precipitation on July 24, 2016, had a $\delta^{18}O$ value of -8.11 ‰ and $\delta^2H$ value of -62.97 ‰, which were smaller than the respective values of pre-event BW (-1.24 ± 0.87 ‰ for $\delta^{18}O$ and -37.79 ± 2.81 ‰ for $\delta^2H$) (Fig. 4). The irrigation water—with a $\delta^{18}O$ of -9.40 ± 0.05 ‰ and $\delta^2H$ of 49.87 ± 2.7 ‰ on August 26, 2016—had a lower $\delta^{18}O$, but a much higher $\delta^2H$ than the pre-irrigation BW (-0.27 ± 0.56 ‰ for $\delta^{18}O$

and -39.21 ± 2.81 ‰ for $\delta^2$H). In summary, the event water in Period I was more depleted in heavy

isotopes than in pre-event BW ($p < 0.05$). In Period II, the event water had a lower $\delta^{18}$O but a higher $\delta^2$H

than pre-event BW ($p < 0.05$).

As expected, the $\delta^2$H and $\delta^{18}$O in BW increased as evaporation occurred during Period I ($p < 0.05$). The

355 increase in $\delta^2$H and $\delta^{18}$O in BW had a significant linear relationship with evaporation time ($p < 0.05$; Fig.

5), suggesting that evaporation favored the lighter water isotopes from BW, resulting in greater $\delta^2$H and

$\delta^{18}$O in BW. In Period II, BW $\delta^{18}$O also increased as evaporation progressed ($p < 0.05$). The increase in

BW $\delta^{18}$O also had a significant linear relationship with evaporation time ($p < 0.05$; Fig. 5). In contrast,

$\delta^2$H of BW decreased linearly with evaporation ($p < 0.01$) in Period II. The slope and intercept both

significantly differed from zero ($p < 0.01$), suggesting that in Period II, evaporation takes away the lighter

O isotope and heavier H isotope from BW.

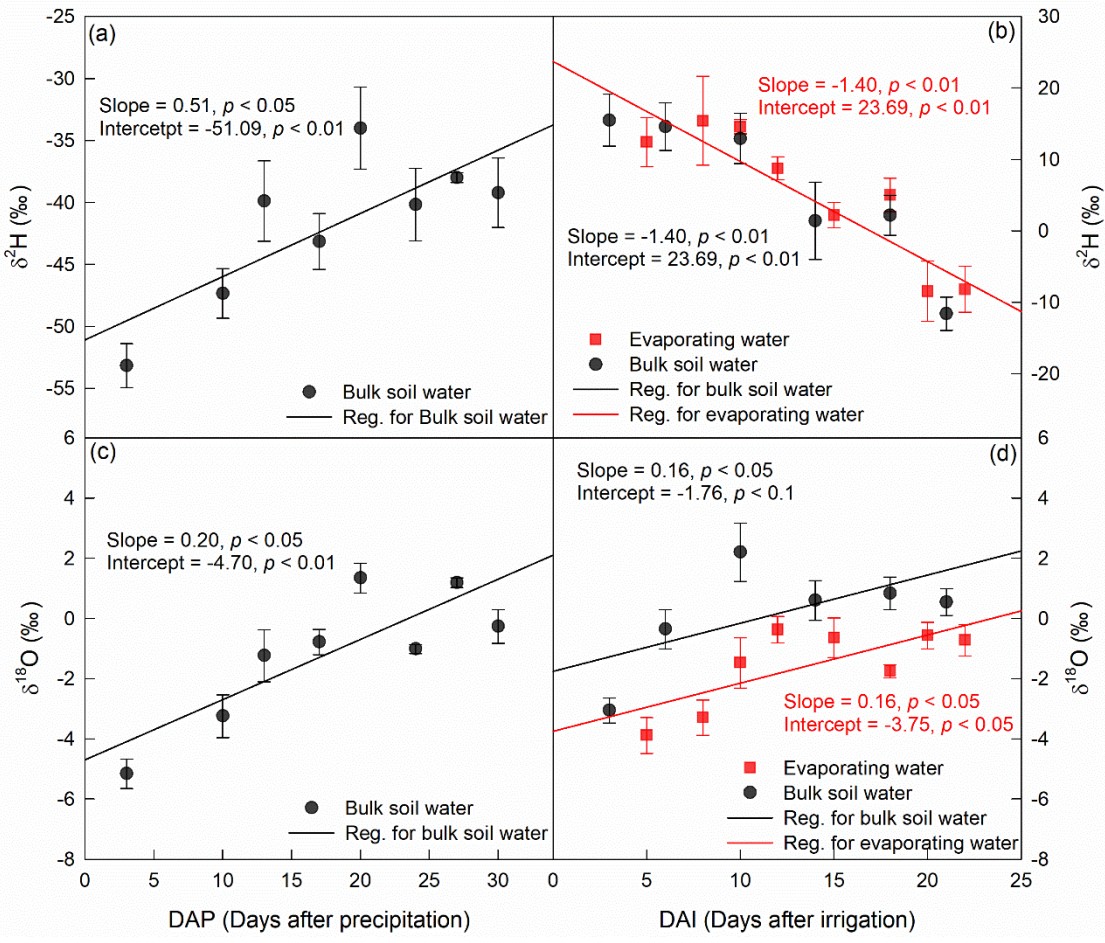

**Figure 5: Temporal variation of $\delta^2$H (upper panel) and $\delta^{18}$O (lower panel) in 0–5-cm bulk soil water and**

**evaporating water during Period I (left column) and Period II (right column). The precipitation occurred on**

**July 24, 2016, and the irrigation took place on August 26, 2016.**

The evaporation line, defined as the change in water isotopes with evaporation time in EW, was remarkably similar to that for BW (Fig. 5). For example, in Period II, $\delta^2H$ in both EW and BW decreased as evaporation proceeded, and both lines had a slope significantly smaller than zero ($p < 0.05$; Fig. 5b). This is contrary to our understanding that evaporation enriches $^2H$ in EW and BW. Moreover, it seemed that EW had higher $^2H$ vales than BW, but the slope and intercept of the EW evaporation line did not differ from that of the BW evaporation line ($p > 0.05$; Fig. 5b).

In period II, $\delta^{18}O$ in both EW and BW increased with evaporation time (Fig. 5d), and the slopes and intercepts significantly differed from zero ($p < 0.05$), indicating that evaporation, as expected, significantly enriched $^{18}O$ in EW and BW. However, there were some differences between EW and BW; $\delta^{18}O$ was consistently more depleted in EW than in BW during this period. Further regression analyses of $\delta^{18}O$ vs. time relationships in EW and BW in Period II indicated that though $\delta^{18}O$ vs. time in EW had the same slope as that in BW ($p > 0.05$), it had significantly smaller intercept than BW ($p < 0.05$). Thus, the linear relationship in $\delta^{18}O$ between EW and BW was given as $\delta^{18}O(EW) = \delta^{18}O(BW)-1.99$ (Fig. 5d). As is well known, the evaporation line ($\delta^{18}O$ vs. time) reflects the evaporative demand and the source water isotopic signature. First, the slopes of the evaporation lines represent the evaporative demand of the atmosphere. Given that EW and BW are under the same evaporative demand, their evaporation lines should have identical slopes. Second, the intercept of the evaporation line represents the isotopic signature of the initial evaporation water source. Therefore, in Period II, the intercepts of an $\delta^{18}O$ value of -1.76 ‰ for BW and -3.75 ‰ for EW represent the initial water sources of BW and EW, respectively. In other words, the sources of water for BW and EW had different isotopic compositions during Period II.

In Period I, we compared the mean $\delta^2H$ and $\delta^{18}O$ values of all measurements within the green circle (Fig. 4) for both EW and BW. The mean $\delta^2H$ and $\delta^{18}O$ values for EW were significantly lower than those for BW ($p < 0.05$). Unfortunately, there were only four data points for EW, so we could not obtain a reliable isotopic relationship between EW and BW.

### 3.3 Variation of deep soil water content, $\delta^2H$, $\delta^{18}O$, and lc-excess

The precipitation event on July 24, 2016, increased the soil water content in the top 60 cm and decreased soil water $\delta^2H$ and $\delta^{18}O$ in the top 20 cm (Fig. 6, upper panel). Therefore, the top 20 cm lc-excess increased at 10 DAP. However, precipitation did not influence the deeper soil $\delta^2H$, $\delta^{18}O$, and lc-excess.

At the end of evaporation Period I (24 DAP), the soil water content decreased in the top 60 cm. In the top 10 cm, soil water $\delta^2H$ and $\delta^{18}O$ increased, and lc-excess decreased.

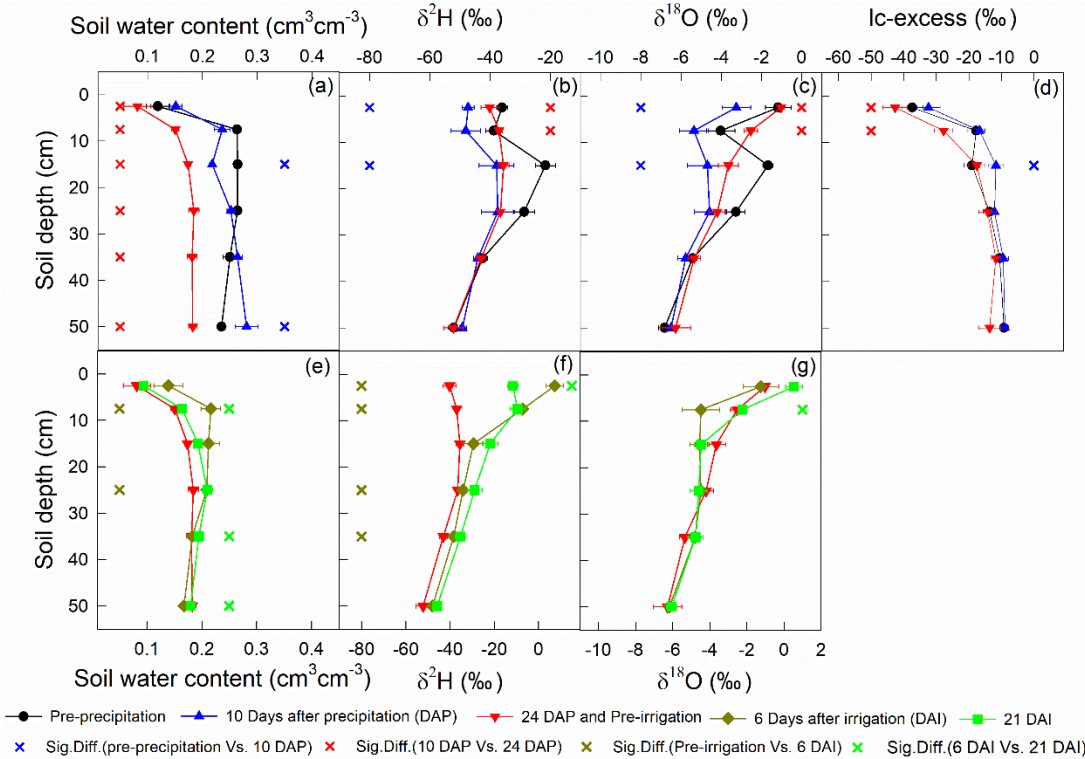

**Figure 6: Temporal variation of deep soil water content, $\delta^2H$, $\delta^{18}O$, and lc-excess during Period I (upper panel)**
**and Period II (lower panel). The precipitation event occurred on July 24, 2016, and the irrigation took place**
**on August 26, 2016.**

Similar to precipitation on July 24, 2016, the irrigation on August 26, 2016, increased the soil water content and decreased the $\delta^{18}O$ of the top 10-cm soil (Fig. 6, lower panel). However, the irrigation event
increased the $\delta^2H$ in the top 20 cm. At the end of evaporation Period II, i.e., 21 DAI, the top 10-cm soil water $\delta^{18}O$ became more enriched whereas $\delta^2H$ became more depleted. Note that the $\delta^2H$ at 5–10 cm was similar to that at 0–5 cm (Fig. 6f).

**3.4 Evaporative water loss derived from bulk soil water and evaporating water**

In Period I, evaporative water loss ($f$) derived from either $\delta^2H$ or $\delta^{18}O$ in BW increased with increasing
evaporation time ($p < 0.01$), and there was no significant difference between them with the same slope and similar intercepts ($p > 0.05$, Fig. 7). The average $f$ values during the period were $0.27 \pm 0.004$ and

0.23 ± 0.002 for $\delta^2H$ and $\delta^{18}O$, respectively. In Period II, $f$ derived from $\delta^{18}O$ in BW and EW increased with evaporation time ($p < 0.05$), and there was no significant difference between them with the same slope and similar intercepts ($p > 0.05$). The average $f$ was 0.27 ± 0.01 and 0.24 ± 0.01 for BW and EW, respectively. However, the evaporative water loss could not be calculated from $\delta^2H$ in BW or EW, as $\delta^2H$ decreased as evaporation progressed (Fig. 5), which was inconsistent with the evaporation theory that soil evaporation enriches heavier water isotopes in the residual soil water. Moreover, we could not calculate the evaporative water loss based on the isotopic composition of EW in Period I, as a reliable linear isotopic relationship between EW and BW could not be obtained from the four data points we had during the period.

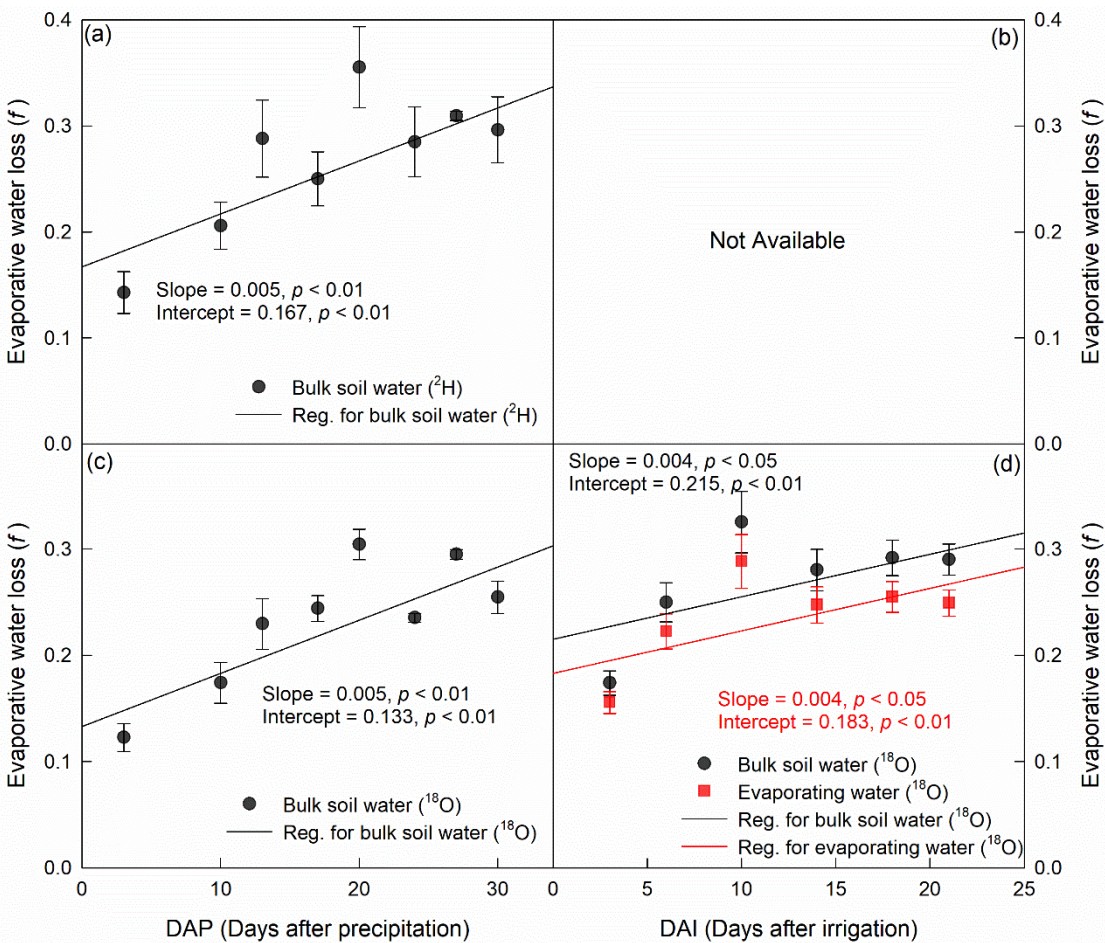

**Figure 7: Temporal variation of evaporative water loss ($f$) derived from isotope value ($\delta^2H$ for upper panel and $\delta^{18}O$ for lower panel) in bulk soil water and evaporating water during Period I (left column) and Period II (right column). The precipitation and irrigation events occurred on July 24, 2016, and August 26, 2016, respectively.**

**4 Discussion**

**4.1 Why evaporating and bulk soil water have different isotopic compositions**

During evaporation, light isotopes are preferentially evaporated, enriching the residual liquid water in heavy isotopes (Mook and De Vries, 2000). This could explain why, with increasing evaporation time, $\delta^2H$ and $\delta^{18}O$ in BW increased in Period I. In Period II, $\delta^{18}O$ (Fig. 5) displayed a similar, increasing trend, whereas $\delta^2H$ had an opposite, decreasing trend. The progressive decrease in $\delta^2H$ with increasing evaporation time cannot be explained by the general notion that with evaporation, residual soil water becomes more enriched with heavy water isotopes. Therefore, there must be a mechanism that preferentially removes $^2H$ or dilutes $^2H$ with $^2H$-depleted water.

For the latter, because there is negligible water input from the atmosphere (both in vapor and liquid form), the only water input could be from the soil below 5 cm. Indeed, because the evaporation amount was larger than the 0–5-cm soil water storage reduction (Section 3.1), the water below 5 cm must have moved upward as evaporation occurred. Consequently, due to evaporation, the order of $\delta^2H$ value should be 0–5 cm > the mixture of pre-evaporation 0–5 cm and 5–10 cm soil water > 5–10 cm. However, 0–5-cm $\delta^2H$ at the end of the evaporation period (21 DAI) was similar to 5–10-cm $\delta^2H$ (Fig. 6f). Moreover, if dilution occurred, the $\delta^{18}O$ would also be diluted, which is not supported by the progressive increase in BW $\delta^{18}O$ during evaporation in the same period and of both $\delta^2H$ and $\delta^{18}O$ in BW of Period I, which should have a deeper soil water contribution (Section 3.1). Therefore, dilution does not substantially affect the isotopic signature of BW. This is further supported by the larger $\delta^{18}O$ in BW in Period II than that in EW (Figs. 4, 5). By deduction, the possible cause of the depletion in $^2H$ would be the preferential removal of $^2H$ from the top 5 cm of soil.

No significant $\delta^2H$ differences were observed between EW and BW in Period II (Fig. 5). However, there was a significant $\delta^{18}O$ difference between EW and BW in Period II, and both $\delta^2H$ and $\delta^{18}O$ in EW differed from the respective values in BW in Period I (Figs. 4, 5). The different isotopic signatures of BW and EW indicate that the water sources for BW and EW were different. Further, the source of EW is closer to the event water than that of BW. This could be explained by a conceptual model of event water and pre-event water partitioning in the soil (Fig. 8).

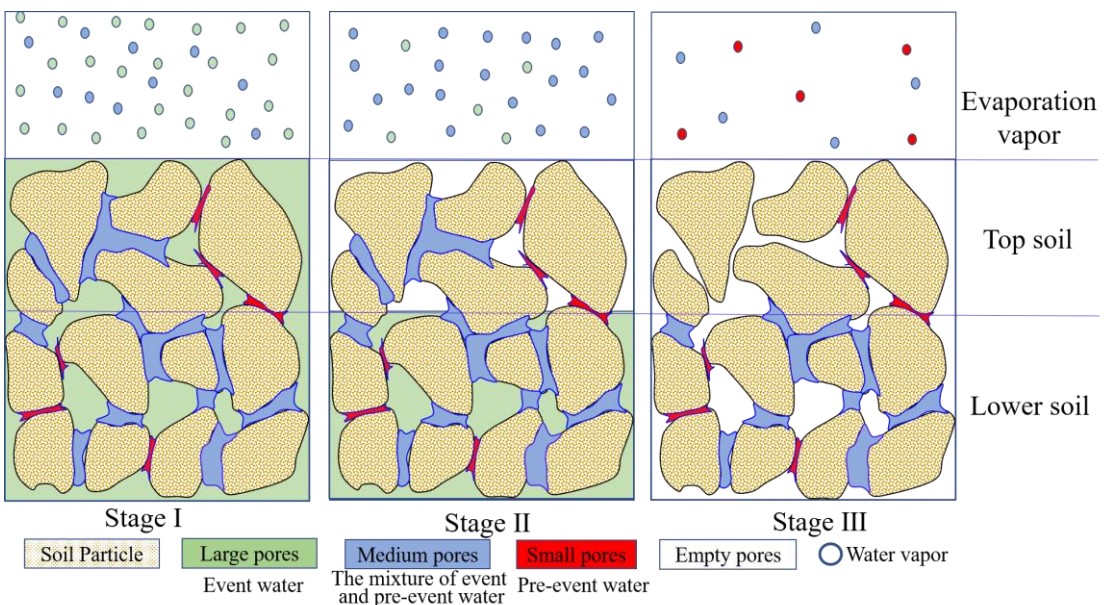

**Figure 8: Schematic of soil pore water partitioning during evaporation.**

### 4.2 Conceptual model for water partitioning in large and small pores during evaporation

For large and intense precipitation events, event water preferentially infiltrates into the empty large pores because of their high hydraulic conductivity. The infiltrated water may partially or fully transfer to the surrounding empty smaller pores, thus bypassing the small soil pores that are filled with pre-event water at the point of water entry and along the infiltration pathway (Beven and Germann, 1982; Booltink and Bouma, 1991; Šimůnek and van Genuchten, 2008; Weiler and Naef, 2003; Zhang et al., 2019). The bypass flow occurs universally (Lin 2010) and has also been reported in our experimental site, the Chinese Loess Plateau (Xiang et al., 2018; Zhang et al., 2019). In our experiment, the precipitation event on July 24, 2016, was 31 mm with the intensity of 10.3 mm h$^{-1}$, and the irrigation event on August 26, 2016, was 30 mm with the intensity of 30 mm h$^{-1}$, and both were sufficient to initiate bypass flow (> 10 mm h$^{-1}$; Beven and Germann 1982; Kumar et al., 1997). The pre-event soil water content was close to residual water content (Section 3.1), indicating that small pores were prefilled with pre-event water. Thus, it is reasonable to assume that the new water filled large pores, and medium pores were likely filled by a mixture of pre-event and event water. Therefore, water in large pores was similar to the event water and water in the small pores was close to the pre-event water, i.e., old event water (Brooks et al., 2010; Sprenger et al., 2019a).

On the other hand, at the end of the evaporation period, lc-excess of 0–5-cm soil at 24 DAP, which had a lower soil water content than in Period II, was still the smallest compared with deeper soil (Fig. 6d).

Therefore, the evaporation front was in the surface soil during both periods. Accordingly, the evaporation in our experiment was in evaporation stage I or II, as indicated in the Introduction. During evaporation stages I and II, small-pore water does not evaporate (Or and Lehmann, 2019; Zhang et al., 2015), and larger-pore water is the primary source of water for evaporation (Lehmann and Or, 2009; Or et al., 2013). Therefore, EW is mainly from larger-pore water, similar to the event water in isotopic composition; BW contains EW and evaporation-insulated small-pore water, similar to the pre-event water. Compared with pre-event water, event water takes evaporation precedence. Therefore, the sequence of water in the evaporation layer can be analogically summarized as adhering to a "last-in-first-out" rule. Thus, when isotopic composition in the event water was smaller than that in pre-event BW, such as $\delta^2H$ and $\delta^{18}O$ in Period I and $\delta^{18}O$ in Period II, the isotopic composition in EW was smaller than that in BW (Fig. 4). When the event water was enriched in heavy isotopes relative to pre-event BW, such as $\delta^2H$ in Period II, EW should be enriched in $^2H$ compared with BW; however, a more precise analysis is needed.

Furthermore, evaporative enrichment and loss of larger-pore water both affect the temporal variation of $\delta^2H$ and $\delta^{18}O$ in EW and BW. When larger-pore water is depleted in heavy isotopes relative to pre-event water, the isotopic composition of EW and BW increases with time; when larger-pore water is enriched in heavy isotopes relative to pre-event water, the enriched water in larger pores empty first, leaving lighter water molecules in BW, which will decrease the isotopic composition in EW and BW with evaporation time.

**4.3 Why the different isotopic compositions in evaporating water and bulk soil water did not make a difference in estimated evaporative water loss?**

There was a significant difference in the isotopic composition between EW and BW; however, the evaporative water loss derived from EW and BW did not differ ($p > 0.05$). As discussed above, the difference between EW and BW is caused by the small-pore water, which does not experience evaporation. The difference in Period II was 1.99 ‰ for $\delta^{18}O$. Nevertheless, the $\delta^{18}O$ difference between EW and BW was too small to make a difference in the calculated evaporative water loss. However, hypothetically increasing the difference from 1.99 ‰ to 3.40 ‰, resulted in a significant difference in the calculated evaporative water loss ($p < 0.05$). The hypothetically calculated $\delta^{18}O$ difference is highly likely in two adjacent precipitation events, based on the 3 years' precipitation isotope data with the largest difference of 16.46 ‰. Many factors could contribute to the differences in isotopic composition between

EW and BW. The first is the relative amount of small-pore water that did not experience evaporation and its isotopic composition difference with EW. The higher the clay content, the greater the amount of small-pore water for the same bulk soil water content (Van Genuchten, 1980). The second is the amount of event water and its isotopic difference with pre-event water. As such, the greater the temporal isotopic variability in precipitation, and evaporation loss, the greater the isotopic difference between EW and BW. Finally, higher soil cations and clay contents also elevate the isotopic difference between EW and BW, as the cations hydrated water and water absorbed by clay particles undergo isotopic fractionation (Gaj et al., 2017a; Oerter et al., 2014). Therefore, an increased difference in isotopic composition between EW and BW may occur for soils with high clay content and salinity and when the amount and isotopic composition differ greatly between event water and pre-event soil water.

The event water was more enriched in heavy isotopes than pre-event soil water, as shown by our $\delta^2$H result in Period II. However, this rarely occurs in nature. Normally, soil water experiences evaporation and thus has more heavy isotopes than precipitation. Nevertheless, when the sub-cloud evaporation effect in precipitation is strong (Salamalikis et al., 2016), precipitation can have more heavy isotopes than pre-event soil water. In this situation, it is impossible to calculate the evaporation ratio using current theories and methods. New theories, or methods to precisely measure water evaporation are needed in this regard. Larger-pore water, preferred by evaporation, also has a relatively higher matric potential and flows more rapidly, and may thus be preferred by roots and dominate groundwater recharge (Sprenger et al., 2018). In other words, evaporation, transpiration, and groundwater preferentially tap the same pool of water, the water that resides in larger soil pores. This is inconsistent with Brooks et al. (2010), who separated soil water into two water worlds: mobile water, which eventually enters the stream, and tightly bound water used by plants. In our study, soil water content was below field capacity and thus according to Brooks et al. (2010), all water in our soil is "tightly bound water", including the large pore water we discussed above. Therefore, in our study, the larger pore water is still under the field capacity, the water that percolates into streams (groundwater) rather slowly and/or is adsorbed by plant roots, which has broad ecohydrological implications.

**5 Conclusion**

We performed an experiment in two continuous evaporation periods: a relatively depleted water input in

Period I and a more enriched $^2$H and depleted $^{18}$O water input in Period II. We collected condensation water using a newly covered plastic film and subsequently calculated the evaporating water's isotopic composition.

The results showed that $\delta^2$H and $\delta^{18}$O in EW had a similar trend to that in BW. When event water was depleted in heavy isotopes relative to pre-event bulk soil water, isotopic composition in EW and BW

increased with increasing evaporation time ($p < 0.05$), and EW was depleted in heavy isotopes relative to BW ($p < 0.05$). When event water was enriched in heavy isotopes relative to pre-event bulk soil water, the isotopic composition in EW and BW decreased with increasing evaporation time ($p < 0.01$). Moreover, the average evaporative water loss derived from $\delta^{18}$O was $0.27 \pm 0.01$ and $0.24 \pm 0.01$ for BW and EW, respectively. The difference between evaporative water loss was negligible owing to the small difference

in $\delta^{18}$O between EW and BW. As $\delta^2$H in BW and EW decreased with evaporation, evaporative water loss could not be obtained using $\delta^2$H. Our results indicate that although the isotopic composition in BW was significantly different from that in EW, the difference was too small to affect evaporative water loss calculation. However, a larger isotopic difference between the event and pre-event water may do. Our research is important for improving our understanding of soil evaporation processes and using isotopes

to study evaporation fluxes.

**Data availability**

The data that support the findings of this study are provided as Supplement.

**Author contribution**

H. Wang, J. Jin, B. Cui, and B. Si designed the research, prepared and interpreted the data, and wrote the

manuscript. M. Wen offered constructive suggestions for the manuscript. H. Wang and X. Ma conducted the fieldwork.

**Competing interests**

The authors declare that they have no conflict of interest.

**Acknowledgement**

This work was partially funded by the National Natural Science Foundation of China (41630860; 41371233) and Natural Science and Engineering Research Council of Canada (NSERC). We thank the China Scholarship Council (CSC) for providing funds (201806300115) to Hongxiu Wang to pursue her studies at the University of Saskatchewan, Canada. We thank Han Li, Wei Xiang, and Huijie Li for the fruitful discussion.

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
