# Peer review of "Technical note: Evaporating water is different from bulk soil water in $\delta^2$ H and $\delta^{18}$ O and implication for evaporation calculation"

_Hydrology and Earth System Sciences, 2020_

## Referee Comment (RC2)

[referee-annotated manuscript omitted]

---

## Author Comment (AC1)

Dear Reviewer,

Thank you very much for your valuable suggestions on our manuscript entitled "Evaporating water is different from bulk soil water in $\delta^2$H and $\delta^{18}$O". We hope we have the opportunity to modify our manuscript for better presentation and interpretation according to your advice. We also did some corrections based on your recommendations and the detailed response to each comment is provided below.

We are looking forward to receiving your feedback.

Sincerely,

Hongxiu Wang

Jingjing Jin

Bingcheng Si

Xiaojun Ma

Mingyi Wen

The "Technical note: Evaporating water is different from bulk soil water in d2H and d18O" describes an experiment to elucidate hysteresis of water isotopic signals during evaporation. The process described is known, but the experiment nicely shows the concept and the implication for deriving evaporative loss from isotopic signals. It is, however, a pity that the difference in d18O was not high enough to result in significant differences in evaporative loss. In this context, it would be beneficial to add more hypothetical calculations under which conditions (difference pre-event/event water) and soils this process might be important. The latter would strengthen the conclusions. In general, it would have been beneficial to have information on soil texture and eventually matric potential.

Another point that is not addressed yet is that evaporation of heavier water than bulk water evaporation loss cannot be calculated. The authors should comment on whether such replacement of heavier isotope occurs under natural conditions and which effect it could have to calculate evaporative loss for natural

isotope abundances. Another main point concerns the description of the calculations. The equation and variables used should be introduced sequentially. The Figures are appropriate and relevant literature cited. However, the manuscript should be corrected by a native speaker (particularly the first part until Discussion). Finally, the title should be adapted since evaporating water is per se different from bulk soil water, and as such, the title does not reflect the process you seek to investigate.

Response: Thanks for your suggestions. We added the discussion on P21 L432-442 : "The differences, in Period II, was 1.99 ‰ for $\delta^{18}O$. Nevertheless, the difference in $\delta^{18}O$ of EW and BW is too small to make a difference on the calculated evaporative water loss. However, by increasing the difference value with 0.01 increment from 1.99 ‰ to 3.40 ‰, there will be a significant difference in the calculated evaporative water loss. The magnitude of isotopic difference of EW and BW is related to the amount of evaporation-insulated small pores water, the amount of event water, and the isotopic difference of event water and pre-event water. Moreover, the maximum amount of evaporation-insulated small pores water is dependent on soil texture, the higher clay content, the greater water amount in small pores (Van Genuchten, 1980). The clay content of our studied soil is 0.24 g g$^{-1}$. Therefore, more attention is needed when dealing with high clay content soil, and when the event water amount and the difference in isotopic composition between event water and pre-event soil water are large.

On the other hand, more precise analysis is needed when the event water is more enriched in heavy isotopes than pre-event soil water as showed by our $\delta^2H$ result in Period II. However, the more enriched precipitation than soil water is rarely happened in nature. Commonly, soil water suffers from evaporation and has more heavy isotopes than precipitation. Nevertheless, when the sub-cloud evaporation effect in precipitation is strong (Salamalikis et al., 2016), the precipitation will contain more heavy isotopes than old precipitation i.e. pre-event soil water, then more attention is needed under this condition." And we modified our title "The different isotopic compositions in evaporating water and bulk soil water did not make a difference in estimated evaporative water loss"

Van Genuchten, M. T.: A closed-form equation for predicting the hydraulic conductivity of unsaturated soils, Soil Sci Soc Am J, 44, 892-898, doi:10.2136/sssaj1980.03615995004400050002x, 1980.
Salamalikis, V., Argiriou, A. A., and Dotsika, E.: Isotopic modeling of the sub-cloud evaporation effect in precipitation, Sci. Total Environ., 544, 1059-1072, doi: 10.1016/j.scitotenv.2015.11.072, 2016.

L26 Please make clear that this is not a general statement but specific to the conditions of your experiment.

Response: Done. The small isotopic difference was added on P1 L27-29 : "We also compared soil evaporation losses derived from water isotopes of EW and BW. With a small magnitude of isotopic difference in EW and BW, the evaporation losses did not differ significantly (p>0.05)"

L28 Which important implications?

Response: We modified the sentence on P1-2 L29-32 : "Our results have important implications for quantifying evaporation processes with water stable isotopes. We hope our study stimulate more researches on the effect of soil water isotopic partitioning in pore space to soil evaporation under different soil conditions and other eco-hydrological processes."

L41 "occupied" seems not the right term in this context.

Response: We used "filled" instead of "occupied". The detailed information was on P2 L46-48 : "When larger soil pores are filled by water, water in small pores does not participate in evaporation (Or and Lehmann, 2019; Zhang et al., 2015)."

Or, D. and Lehmann, P.: Surface evaporative capacitance: How soil type and rainfall characteristics affect global-scale surface evaporation, Water Resour. Res., 55, 519-539, doi:10.1029/2018WR024050, 2019.

Zhang, C., Li, L., and Lockington, D.: A physically based surface resistance model for evaporation from bare soils, Water Resour. Res., 51, 1084-1111, doi:10.1002/2014wr015490, 2015.

L45 Please rephrase the sentence.

Response: We rephrased the sentence on P2 L50-513 : "With the progressive reduction of water in larger pores, the evaporation rate decreases gradually."

L54 Large pores instead of pore.

Response: Done.

L37-59 This section should be moved to the methods.

Response: Thanks very much for your suggestion. We really appreciate your help. However, in order to have a good logic in the introduction section, we have to keep the evaporation processes in the introduction section. Combining the evaporation process and infiltration process, we raised our scientific question, which is evaporating water should has different isotopic composition from bulk soil

water. Moreover, we hope we can have the opportunity to have a further discussion on it with you.

Thanks.

L82-84 This should be moved to the method section.

Response: Thanks for the suggestion. Done.

L84 Rephrase: This study may help to ….

Response: Thanks. The sentence was rephrased on P3 L87-88 : "This study may help to improve our understanding to the process of soil evaporation and the ecohydrological water cycle."

L98 Add values or signature in the sentence.

Response: Done. The values were added in P4 L99-101 : "The field was irrigated 30 mm of mixed tap water ($\delta^2$H = -61.11 ‰, $\delta^{18}$O = -9.42 ‰) and deuterium enriched water (the $^2$H concentration was 99.96 %, $\delta^2$H = 1.60 × $10^{10}$ ‰, Cambridge Isotope Laboratories, Inc.) on 2016/8/26."

L107 "secondary" evaporation

Response: Done.

L129-130 It is not clear whether the authors refer in this sentence to their own findings (in this case I would move the sentence to the results) or if there refer to other studies (in this case they should be cited). Moreover, the structure of the sentence is not clear and should be corrected.

Response: It is referred to other studies. I separated the citations of the next sentence to two parts. The detailed information was added on P6 L141-146 : "However, in terms of isotopic compositions, the extracted water is depleted in heavy isotope than the reference water and the depletion is related with soil clay contents and water contents due to incomplete soil water extraction (Orlowski et al., 2016; Orlowski et al., 2013). In order to extract all of the water from our soil samples with the clay content of 0.24 g g$^{-1}$, higher temperature (>200 $^o$C) is suggested to be used for soil water extractions (Gaj et al., 2017a; Gaj et al., 2017b; Orlowski et al., 2018)."

Orlowski, N., Breuer, L., and McDonnell, J. J.: Critical issues with cryogenic extraction of soil water for stable isotope analysis, Ecohydrology, 9, 1-5, doi:10.1002/eco.1722, 2016.

Orlowski, N., Frede, H. G., Brüggemann, N., and Breuer, L.: Validation and application of a cryogenic vacuum extraction system for soil and plant water extraction for isotope analysis, J. Sens. Sens. Syst., 2, 179-193, doi:10.5194/jsss-2-179-2013, 2013.

Gaj, M., Kaufhold, S., Koeniger, P., Beyer, M., Weiler, M., and Himmelsbach, T.: Mineral mediated isotope fractionation of soil water, Rapid Commun. Mass Spectrom., 31, 269-280, doi:10.1002/rcm.7787, 2017a.

Gaj, M., Kaufhold, S., and McDonnell, J. J.: Potential limitation of cryogenic vacuum extractions and spiked experiments, Rapid Commun. Mass Spectrom., 31, 821-823, doi: 10.1002/rcm.7850, 2017b.

Orlowski, N., Breuer, L., Angeli, N., Boeckx, P., Brumbt, C., Cook, C. S., ... and McDonnell, J. J.: Interlaboratory comparison of cryogenic water extraction systems for stable isotope analysis of soil water, Hydrol Earth Syst Sci, 22, 3619-3637, doi:10.5194/hess-22-3619-2018, 2018.

L131: When are higher temperature needed? In case of higher clay content. This is not clear from the sentence. Could you provide soil texture information?

Response: Done. The soil texture information was added on P6 L144-146 : "In order to extract all of the water from our soil samples with the clay content of 0.24 g g$^{-1}$, higher temperature (>200 $^{o}$C) is suggested to be used for soil water extractions (Gaj et al., 2017a; Gaj et al., 2017b; Orlowski et al., 2018)."

Gaj, M., Kaufhold, S., Koeniger, P., Beyer, M., Weiler, M., and Himmelsbach, T.: Mineral mediated isotope fractionation of soil water, Rapid Commun. Mass Spectrom., 31, 269-280, doi:10.1002/rcm.7787, 2017a.

Gaj, M., Kaufhold, S., and McDonnell, J. J.: Potential limitation of cryogenic vacuum extractions and spiked experiments, Rapid Commun. Mass Spectrom., 31, 821-823, doi: 10.1002/rcm.7850, 2017b.

Orlowski, N., Breuer, L., Angeli, N., Boeckx, P., Brumbt, C., Cook, C. S., ... and McDonnell, J. J.: Interlaboratory comparison of cryogenic water extraction systems for stable isotope analysis of soil water, Hydrol Earth Syst Sci, 22, 3619-3637, doi:10.5194/hess-22-3619-2018, 2018.

L147: "sub samples"

Response: Done.

L159 Use the present tense for referring to Tables and Figures.

Response: Done.

L167- : Why did you change the soil of the lysimeters. The reason is not apparent.

Response: The reason to change the soil of the lysimeters was added on P7 L184-185 : "Further, in order to keep better representative of the field soil, the soil of the inside lysimeter was changed every four days."

L222-228: Here, the introduction of the variables and equations is mixed up and difficult to follow. Please introduce each equation with its variables from top to bottom since this is an important aspect of your study.

Response: Thanks for pointing out our mistakes. We rewrote the introduction of the variables and equations and the detailed information can be found on P10 L232-250 : "The evaporative water losses were estimated using Eqs. (10-16) (Hamilton et al., 2005; Skrzypek et al., 2015; Sprenger et al., 2017), which is based on bulk soil water isotope balance and Craig-Gordon model.

$$f = 1 - \left[\frac{\delta_{BW} - \delta^*}{\delta_I - \delta^*}\right]^{\frac{1}{m}} , \tag{10}$$

where f represents the ratio of evaporative water loss to the total water source; $\delta_{BW}$ is the isotopic signal of 0-5 cm bulk soil water; $\delta_I$ is defined as the isotopic signal of the original water source by calculating the intercept between the evaporation line of the 0-5 cm bulk soil water isotope data in Period I in the dual-isotope plot and the LMWL (Fig. 3); $m$ and $\delta^*$ are described blow.

$$m = \frac{h - \frac{\varepsilon}{1000}}{1 - h + \frac{\varepsilon_k}{1000}} , \tag{11}$$

$$\delta^* = \frac{h * \delta_A + \varepsilon}{h - \frac{\varepsilon}{1000}} , \tag{12}$$

where $h$ is the average ambient air relative humidity over 30 days prior to each soil water sampling (Sprenger et al., 2017); $\varepsilon$ is the total enrichment factor; $\varepsilon_k$ is the kinetic enrichment factor; $\delta_A$ is the ambient vapor isotopic composition.

$$\varepsilon = (1 - \alpha_1^*) * 1000 + \varepsilon_k , \tag{13}$$

$$\varepsilon_k(18O) = 28.5(1 - h) , \tag{14}$$

$$\varepsilon_k(2H) = 25.115(1 - h) , \tag{15}$$

$$\delta_A = (\delta_{rain} - (\alpha_A^+ - 1) * 1000)/\alpha_A^+ , \tag{16}$$

where $\alpha_1^*$ is the equilibrium fractionation factor under the field soil condition (Fig. 1); $\alpha_A^+$ is the equilibrium fractionation factor in the ambient air; $\delta_{rain}$ is the amount weighted isotopic composition in precipitation from 2016/7/11 to 2016/9/16."

Hamilton, S. K., Bunn, S. E., Thoms, M. C., and Marshall, J. C.: Persistence of aquatic refugia between flow pulses in a dryland river system (Cooper Creek, Australia), Limnol. Oceanogr., 50, 743-754, doi:10.4319/lo.2005.50.3.0743, 2005.

Skrzypek, G., Mydłowski, A., Dogramaci, S., Hedley, P., Gibson, J. J., and Grierson, P. F.: Estimation of evaporative loss based on the stable isotope composition of water using Hydrocalculator, J. Hydrol., 523, 781-789, doi:10.1016/j.jhydrol.2015.02.010, 2015.

Sprenger, M., Tetzlaff, D., and Soulsby, C.: Soil water stable isotopes reveal evaporation dynamics at the soil–plant–atmosphere interface of the critical zone, Hydrol Earth Syst Sci, doi:10.5194/hess-21-3839-2017, 2017.

L240: Is the variable n introduced?

Response: Done. The introduction of variable n was added on P11 L261-263 :

$$``f = 1 - \left[\frac{\delta_{BW} - \delta^* + n}{\delta_I - \delta^* + n}\right]^{\frac{1}{m}}, \tag{17}$$

where n is an intermediate variable and can be expressed as following,

$$n = \frac{-2.13\alpha_1^*}{h - \frac{\varepsilon}{1000}}, \tag{18}"$$

L242. The article is missing: A general linear …

Response: Done.

L270: Delete "was".

Response: Done.

L277: What is meant by newly added water? The irrigation water? Please use the same terminology as before.

Response: Thanks. We used event water to describe precipitation event water and irrigation event water. All the manuscript was modified.

L300-306: style: delete some "therefore"

Response: Done.

L414: Could you explain how you estimated the value of 3.52 to result in significant differences.

Response: the method was added on P21 L434-436 : "However, by increasing the difference value with 0.01 increment from 1.99 ‰ to 3.40 ‰, there will be a significant difference in the calculated evaporative water loss."

L418: Do you mean matric potential?

Response: Yes, you are right. We changed it to matric potential on P21 L449-450 : "While evaporation prefers larger pores water, larger pores water also has relative higher matric potential and therefore, may also be preferred by roots and dominate groundwater recharge (Sprenger et al., 2018)"

Sprenger, M., Tetzlaff, D., Buttle, J., Laudon, H., and Soulsby, C.: Water ages in the critical zone of long-term experimental sites in northern latitudes, Hydrol Earth Syst Sci, doi:10.5194/hess-22-3965-2018, 2018.

L436: Please make clear that this statement refers only to small differences in isotopic signals.

Response: Done. The information was added on P22 L467-470 : "Our results indicate that even isotopic

composition in BW is significantly different from that in EW, the small difference does not affect evaporative water loss calculation. However, more attention is needed when there is a large isotopic difference between event water and pre-event water."

---

## Author Response (AR1)

Dear Dr. Orlowski,

Thank you very much for the valuable suggestions on our manuscript entitled "Evaporating water is different from bulk soil water in $\delta^2H$ and $\delta^{18}O$ and implication for evaporation calculation". Corrections have been made based on your recommendations, and the detailed response to each comment is provided below. We tracked all the corrections within the revised manuscript and noted the new line numbers in this document where the corrections can be found in the revised manuscript.

We are looking forward to receiving your feedback.

Sincerely,

Hongxiu Wang

Jingjing Jin

Bingcheng Si

Xiaojun Ma

Mingyi Wen

**Comments from Editor**

Thank you very much for submitting your responses to the reviewer's comments. When resubmitting your revised manuscript version,

(1) please consider improving the presentation and interpretation of the results (following ref. 2), improving the figure quality, labeling and description

Response: We tried our best to improve the figure quality (a larger font size was used for the axis labels of all the figures), labeling, and description, and the presentation and interpretation of the results.

(2) and include a stronger soil physical view on your data.

Response: For the soil physical view, we added the field capacity and residual water content information on P6 L135-145: "To obtain bulk soil density, field capacity, and residual water content, three 70-cm deep pits were dug at the end of the growing season. Stainless rings with a volume of 100 cm$^3$ (DIK-

1801; Daiki Rika Kogyo Co., Ltd, Saitama, Japan) were pushed into the face of each soil pit at depths of 10, 20, 40, and 60 cm to obtain the soil samples. The soil samples were then saturated with distilled water, weighed, and placed in a high-speed centrifuge (CR21GII; Hitachi, Tokyo, Japan) with a centrifugation rotation velocity equivalent to a soil suction of 1 kPa for 10 min. The soil samples were weighed again to obtain the gravimetric water content at the aforementioned suction. This was repeated for suctions of 5, 10, 30, 50, 70, 100, 300, 500, and 700 kPa for 17, 26, 42, 49, 53, 58, 73, 81, and 85 min, respectively, to obtain the soil characteristic curve. After centrifugation, the soil samples were oven-dried and weighed to obtain the bulk soil density, which was used to convert gravimetric water content to volumetric water content."

(3) As suggested by ref 1, adding more hypothetical calculations for evaporative loss effects on $^{18}O$ would enhance the quality of the manuscript.

Response: The hypothetical calculations for evaporative water loss effects on $^{18}O$ was added on P22 L481-486: "The difference in Period II was 1.99 ‰ for $\delta^{18}O$. Nevertheless, the $\delta^{18}O$ difference between EW and BW was too small to make a difference in the calculated evaporative water loss. However, hypothetically increasing the difference from 1.99 ‰ to 3.40 ‰, resulted in a significant difference in the calculated evaporative water loss ($p < 0.05$). The hypothetically calculated $\delta^{18}O$ difference is highly likely in two adjacent precipitation events, based on the 3 years' precipitation isotope data with the largest difference of 16.46 ‰."

(4) The manuscript would further benefit from a discussion on soil physio-chemical properties and their effects on the soil's isotopic composition.

Response: For soil physio-chemical properties and their effects on the soil's isotopic composition, we added the discussion on P22 L486-496: "Many factors could contribute to the differences in isotopic composition between EW and BW. The first is the relative amount of small-pore water that did not experience evaporation and its isotopic composition difference with EW. The higher the clay content, the greater the amount of small-pore water for the same bulk soil water content (Van Genuchten, 1980). The second is the amount of event water and its isotopic difference with pre-event water. As such, the greater the temporal isotopic variability in precipitation, and evaporation loss, the greater the isotopic difference between EW and BW. Finally, higher soil cations and clay contents also elevate the isotopic difference

between EW and BW, as the cations hydrated water and water absorbed by clay particles undergo isotopic fractionation (Gaj et al., 2017a; Oerter et al., 2014). Therefore, an increased difference in isotopic composition between EW and BW may occur for soils with high clay content and salinity and when the amount and isotopic composition differ greatly between event water and pre-event soil water."

We also add more soil physics on evaporation processes on P2 L46-62: "Water loss from soil progresses with air invasion into the soil in the order of large to small pores (Aminzadeh and Or, 2014; Lehmann and Or, 2009; Or et al., 2013). Soil pores can be divided into large, medium, and small pores. There is a minimum amount of small pore water at which liquid water in soil is still continuous or connected, below which liquid water is no longer connected, and vapor transport is the only way to further reduce water in soil. This water content is called the residual water content in the soil characteristic curve (Van Genuchten, 1980; Zhang et al., 2015). When large soil pores are filled with water, water in small pores does not participate in evaporation (Or and Lehmann, 2019; Zhang et al., 2015). Therefore, soil evaporation can be divided into three stages (Hillel, 1998; Or et al, 2013). Stage I: the evaporation front is in the surface soil, and water in large and medium pores participates in evaporation, but larger pores are the primary contributors. With the progressive reduction of water in the larger pores, the evaporation rate gradually decreases. Stage II: evaporation front is still in the surface soil, but larger pores are filled with air, water residing in the medium soil pores in the surface soil evaporates, and deep larger soil pores recharge the surface medium pores by capillary pull (Or and Lehmann, 2019), and the evaporation rate remains constant. Stage III: the hydraulic connectivity between the surface medium pores and deep large pores breaks, such that the evaporation front recedes into the subsurface soil. Water in the surface small pores and water in medium pores on the evaporation front evaporates. The evaporation rate decreases to a low value (Or et al, 2013)."

Van Genuchten, M. T.: A closed-form equation for predicting the hydraulic conductivity of unsaturated soils, Soil Sci Soc Am J, 44, 892-898, doi:10.2136/sssaj1980.03615995004400050002x, 1980.

Gaj, M., Kaufhold, S., Koeniger, P., Beyer, M., Weiler, M., and Himmelsbach, T.: Mineral mediated isotope fractionation of soil water, Rapid Commun. Mass Spectrom., 31, 269-280, doi:10.1002/rcm.7787, 2017a.

Oerter, E., Finstad, K., Schaefer, J., Goldsmith, G. R., Dawson, T., and Amundson, R.: Oxygen isotope fractionation effects in soil water via interaction with cations (Mg, Ca, K, Na) adsorbed to phyllosilicate clay minerals, J. Hydrol., 515, 1-9, doi:10.1016/j.jhydrol.2014.04.029, 2014.

Aminzadeh, M. and Or, D.: Energy partitioning dynamics of drying terrestrial surfaces, J. Hydrol., 519, 1257-1270, doi:10.1016/j.jhydrol.2014.08.037, 2014.

Lehmann, P. and Or, D.: Evaporation and capillary coupling across vertical textural contrasts in porous media, Phys. Rev. E, 80, 046318, doi:10.1103/PhysRevE.80.046318, 2009.

Or, D., Lehmann, P., Shahraeeni, E., and Shokri, N.: Advances in soil evaporation physics—A review, Vadose Zone J, 12, 1-16, doi:10.2136/vzj2012.0163, 2013.

Zhang, C., Li, L., and Lockington, D.: A physically based surface resistance model for evaporation from bare soils, Water Resour. Res., 51, 1084-1111, doi:10.1002/2014wr015490, 2015.

Or, D. and Lehmann, P.: Surface evaporative capacitance: How soil type and rainfall characteristics affect global-scale surface evaporation, Water Resour. Res., 55, 519-539, doi:10.1029/2018WR024050, 2019.

Hillel, D.: Environmental soil physics: Fundamentals, applications, and environmental considerations, Elsevier, 1998.

(5) In line with ref 1, I would suggest introducing the equations and variables in a better way.

Response: We rewrote the introduction of the variables and equations and the detailed information can be found on P10 L258-287: "For an open system (field soil condition, Fig. 1c), evaporation from surface soil water to ambient air undergoes two processes: the equilibrium fractionation process from the surface soil to the saturated vapor layer above the soil surface and the kinetic fractionation process from the saturated vapor layer to ambient air. The isotopic composition of evaporation vapor is controlled by the isotope values of the evaporating soil water and ambient vapor, equilibrium, and kinetic fractionations. The kinetic fractionation can be described by the enrichment factors ($\varepsilon_k$) of $^{18}O$ and $^2H$ as a function of ambient air relative humidity ($h$) (Gat 1996):

$$\varepsilon_k(^{18}O) = 28.5(1-h), \tag{10}$$

$$\varepsilon_k(^2H) = 25.115(1-h), \tag{11}$$

The total enrichment factor, $\varepsilon$, can be obtained from the kinetic enrichment factor ($\varepsilon_k$) and equilibrium fractionation factor ($\alpha_3^*$) (Skrzypek et al., 2015):

$$\varepsilon = (1-\alpha_3^*)*1000 + \varepsilon_k, \tag{12}$$

The ambient vapor isotopic composition ($\delta_A$) can be obtained as follows (Gibson et al., 2008):

$$\delta_A = (\delta_{rain} - (\alpha_A^+ - 1)*1000)/\alpha_A^+, \tag{13}$$

where $\alpha_A^+$ is the equilibrium fractionation factor in the ambient air, $\delta_{rain}$ is the amount weighted isotopic composition in precipitation from July 11, to September 16, 2016.

The isotopic compositions of bulk soil water and evaporating water can be used to evaporating soil water in the Craig-Gordon model (Eq. 14) to calculate the isotope value of the evaporation vapor ($\delta_{EV}$).

$$\delta_{EV} = \frac{\alpha_3^* \delta_{BW} - h\delta_A - \varepsilon}{(1-h) + \varepsilon_k/1000} \text{ or } \frac{\alpha_3^* \delta_{EW} - h\delta_A - \varepsilon}{(1-h) + \varepsilon_k/1000} \tag{14}$$

Based on the bulk soil water isotope mass balance, i.e., the change in bulk soil water isotopic composition multiplied by the soil water reduction equals the evaporation vapor isotopic composition multiplied by the evaporation amount (Hamilton et al., 2005; Skrzypek et al., 2015; Sprenger et al., 2017), we can calculate evaporative water loss to the total water source ($f$).

$$f = 1 - \left[\frac{\delta_{BW} - \delta^*}{\delta_I - \delta^*}\right]^{\frac{1}{m}}, \tag{15}$$

where $\delta_I$ is the isotopic signal of the original water source. $\delta_I$ is generally unknown and can be conveniently obtained by calculating the intersection between the regression line of the 0–5-cm bulk soil water isotope in Period I and the LMWL in the dual-isotope plot (Fig. 3). $m$ and $\delta^*$ in Eq. (15) are given by:

$$m = \frac{h - \frac{\varepsilon}{1000}}{1 - h + \frac{\varepsilon_k}{1000}}, \tag{16}$$

$$\delta^* = \frac{h * \delta_A + \varepsilon}{h - \frac{\varepsilon}{1000}}, \tag{17}$$

"

Gat J R.: OXYGEN AND HYDROGEN ISOTOPES IN THE HYDROLOGIC CYCLE. Annu. rev. earth. planet. sci, 24, 225-262, doi:10.1146/annurev.earth.24.1.225, 1996.

Gibson, J. J., Birks, S. J., and Edwards, T.: Global prediction of da and d2h-d18o evaporation slopes for lakes and soil water accounting for seasonality. Global Biogeochem. Cy., 22, doi:10.1029/2007GB002997, 2008.

Hamilton, S. K., Bunn, S. E., Thoms, M. C., and Marshall, J. C.: Persistence of aquatic refugia between flow pulses in a dryland river system (Cooper Creek, Australia), Limnol. Oceanogr., 50, 743-754, doi:10.4319/lo.2005.50.3.0743, 2005.

Skrzypek, G., Mydłowski, A., Dogramaci, S., Hedley, P., Gibson, J. J., and Grierson, P. F.: Estimation of evaporative loss based on the stable isotope composition of water using Hydrocalculator, J. Hydrol., 523, 781-789, doi:10.1016/j.jhydrol.2015.02.010, 2015.

Sprenger, M., Tetzlaff, D., and Soulsby, C.: Soil water stable isotopes reveal evaporation dynamics at the soil–plant–atmosphere interface of the critical zone, Hydrol Earth Syst Sci, doi:10.5194/hess-21-3839-2017, 2017.

(6) I would recommend an English proofreading of the manuscript by a native speaker.

Response: Thanks for your suggestion. Our paper is professionally edited by the Elsevier Language Editing Services.

**Comments from Reviewer: 1**

**Main comment:** The "Technical note: Evaporating water is different from bulk soil water in $\delta^2H$ and $\delta^{18}O$" describes an experiment to elucidate hysteresis of water isotopic signals during evaporation. The process described is known, but the experiment nicely shows the concept and the implication for deriving evaporative loss from isotopic signals. It is, however, a pity that the difference in $\delta^{18}O$ was not high enough to result in significant differences in evaporative loss. In this context,

(1) it would be beneficial to add more hypothetical calculations under which conditions (difference pre-event/event water) and soils this process might be important. The latter would strengthen the conclusions.

Response: To address more hypothetical calculations for evaporative water loss effects on $^{18}O$, we added the discussion on P22 L481-496: "The difference in Period II was 1.99 ‰ for $\delta^{18}O$. Nevertheless, the $\delta^{18}O$ difference between EW and BW was too small to make a difference in the calculated evaporative water loss. However, hypothetically increasing the difference from 1.99 ‰ to 3.40 ‰, resulted in a significant difference in the calculated evaporative water loss ($p < 0.05$). The hypothetically calculated $\delta^{18}O$ difference is highly likely in two adjacent precipitation events, based on the 3 years' precipitation isotope data with the largest difference of 16.46 ‰. Many factors could contribute to the differences in isotopic composition between EW and BW. The first is the relative amount of small-pore water that did not experience evaporation and its isotopic composition difference with EW. The higher the clay content, the greater the amount of small-pore water for the same bulk soil water content (Van Genuchten, 1980). The second is the amount of event water and its isotopic difference with pre-event water. As such, the greater the temporal isotopic variability in precipitation, and evaporation loss, the greater the isotopic difference between EW and BW. Finally, higher soil cations and clay contents also elevate the isotopic difference between EW and BW, as the cations hydrated water and water absorbed by clay particles undergo isotopic fractionation (Gaj et al., 2017a; Oerter et al., 2014). Therefore, an increased difference in isotopic composition between EW and BW may occur for soils with high clay content and salinity and when the amount and isotopic composition differ greatly between event water and pre-event soil water"

Van Genuchten, M. T.: A closed-form equation for predicting the hydraulic conductivity of unsaturated soils, Soil Sci Soc Am J, 44, 892-898, doi:10.2136/sssaj1980.03615995004400050002x, 1980.

Gaj, M., Kaufhold, S., Koeniger, P., Beyer, M., Weiler, M., and Himmelsbach, T.: Mineral mediated isotope fractionation of soil water, Rapid Commun. Mass Spectrom., 31, 269-280,

doi:10.1002/rcm.7787, 2017a.

Oerter, E., Finstad, K., Schaefer, J., Goldsmith, G. R., Dawson, T., and Amundson, R.: Oxygen isotope fractionation effects in soil water via interaction with cations (Mg, Ca, K, Na) adsorbed to phyllosilicate clay minerals, J. Hydrol., 515, 1-9, doi:10.1016/j.jhydrol.2014.04.029, 2014.

(2) In general, it would have been beneficial to have information on soil texture and eventually matric potential.

Response: The soil texture information was added on P6 L155-157: "To extract all water from a soil sample, a higher extraction temperature (>200 °C) might be desirable, especially for soils with substantial clay particles such as in the present study (clay content of 0.24 g g$^{-1}$) (Gaj et al., 2017a; Gaj et al., 2017b; Orlowski et al., 2018)." As soil matric potential is related to soil pore size, and the smaller size of pores, the lower matric potential. Further, the amount of small pores is determined by clay content. Therefore, we added the discussion about clay content and small pores on P22 L486-489: "Many factors could contribute to the differences in isotopic composition between EW and BW. The first is the relative amount of small-pore water that did not experience evaporation and its isotopic composition difference with EW. The higher the clay content, the greater the amount of small-pore water for the same bulk soil water content (Van Genuchten, 1980)."

Gaj, M., Kaufhold, S., Koeniger, P., Beyer, M., Weiler, M., and Himmelsbach, T.: Mineral mediated isotope fractionation of soil water, Rapid Commun. Mass Spectrom., 31, 269-280, doi:10.1002/rcm.7787, 2017a.

Gaj, M., Kaufhold, S., and McDonnell, J. J.: Potential limitation of cryogenic vacuum extractions and spiked experiments, Rapid Commun. Mass Spectrom., 31, 821-823, doi: 10.1002/rcm.7850, 2017b.

Orlowski, N., Breuer, L., Angeli, N., Boeckx, P., Brumbt, C., Cook, C. S., ... and McDonnell, J. J.: Interlaboratory comparison of cryogenic water extraction systems for stable isotope analysis of soil water, Hydrol Earth Syst Sci, 22, 3619-3637, doi:10.5194/hess-22-3619-2018, 2018.

Van Genuchten, M. T.: A closed-form equation for predicting the hydraulic conductivity of unsaturated soils, Soil Sci Soc Am J, 44, 892-898, doi:10.2136/sssaj1980.03615995004400050002x, 1980.

(3) Another point that is not addressed yet is that evaporation of heavier water than bulk water evaporation loss cannot be calculated. The authors should comment on whether such replacement of heavier isotope occurs under natural conditions and which effect it could have to calculate evaporative loss for natural isotope abundances.

Response: For the heavy isotope enriched precipitation issue, we added the discussion on P23 L497-502: "The event water was more enriched in heavy isotopes than pre-event soil water, as shown by our $\delta^2H$ result in Period II. However, this rarely occurs in nature. Normally, soil water experiences evaporation and thus has more heavy isotopes than precipitation. Nevertheless, when the sub-cloud evaporation effect in precipitation is strong (Salamalikis et al., 2016), precipitation can have more heavy isotopes than pre-event soil water. In this situation, it is impossible to calculate the evaporation ratio using current theories and methods. New theories, or methods to precisely measure water evaporation are needed in this regard."

Salamalikis, V., Argiriou, A. A., and Dotsika, E.: Isotopic modeling of the sub-cloud evaporation effect in precipitation, Sci. Total Environ., 544, 1059-1072, doi: 10.1016/j.scitotenv.2015.11.072, 2016.

(4) Another main point concerns the description of the calculations. The equation and variables used should be introduced sequentially. The Figures are appropriate and relevant literature cited.

Response: We rewrote the introduction of the variables and equations and the detailed information can be found on P10 L258-287: "For an open system (field soil condition, Fig. 1c), evaporation from surface soil water to ambient air undergoes two processes: the equilibrium fractionation process from the surface soil to the saturated vapor layer above the soil surface and the kinetic fractionation process from the saturated vapor layer to ambient air. The isotopic composition of evaporation vapor is controlled by the isotope values of the evaporating soil water and ambient vapor, equilibrium, and kinetic fractionations. The kinetic fractionation can be described by the enrichment factors ($\varepsilon_k$) of $^{18}O$ and $^2H$ as a function of ambient air relative humidity ($h$) (Gat 1996):

$$\varepsilon_k(^{18}O) = 28.5(1-h), \tag{10}$$

$$\varepsilon_k(^2H) = 25.115(1-h), \tag{11}$$

The total enrichment factor, $\varepsilon$, can be obtained from the kinetic enrichment factor ($\varepsilon_k$) and equilibrium fractionation factor ($\alpha_3^*$) (Skrzypek et al., 2015):

$$\varepsilon = (1 - \alpha_3^*) * 1000 + \varepsilon_k, \tag{12}$$

The ambient vapor isotopic composition ($\delta_A$) can be obtained as follows (Gibson et al., 2008):

$$\delta_A = (\delta_{rain} - (\alpha_A^+ - 1) * 1000)/\alpha_A^+ , \tag{13}$$

where $\alpha_A^+$ is the equilibrium fractionation factor in the ambient air, $\delta_{rain}$ is the amount weighted isotopic composition in precipitation from July 11, to September 16, 2016.

The isotopic compositions of bulk soil water and evaporating water can be used to evaporating soil water in the Craig-Gordon model (Eq. 14) to calculate the isotope value of the evaporation vapor ($\delta_{EV}$).

$$\delta_{EV} = \frac{\alpha_3^* \delta_{BW} - h\delta_A - \varepsilon}{(1-h) + \varepsilon_k/1000} \quad or \quad \frac{\alpha_3^* \delta_{EW} - h\delta_A - \varepsilon}{(1-h) + \varepsilon_k/1000} \tag{14}$$

Based on the bulk soil water isotope mass balance, i.e., the change in bulk soil water isotopic composition multiplied by the soil water reduction equals the evaporation vapor isotopic composition multiplied by the evaporation amount (Hamilton et al., 2005; Skrzypek et al., 2015; Sprenger et al., 2017), we can calculate evaporative water loss to the total water source (*f*).

$$f = 1 - \left[\frac{\delta_{BW} - \delta^*}{\delta_I - \delta^*}\right]^{\frac{1}{m}} , \tag{15}$$

where $\delta_I$ is the isotopic signal of the original water source. $\delta_I$ is generally unknown and can be conveniently obtained by calculating the intersection between the regression line of the 0–5-cm bulk soil water isotope in Period I and the LMWL in the dual-isotope plot (Fig. 3). $m$ and $\delta^*$ in Eq. (15) are given by:

$$m = \frac{h - \frac{\varepsilon}{1000}}{1 - h + \frac{\varepsilon_k}{1000}} , \tag{16}$$

$$\delta^* = \frac{h * \delta_A + \varepsilon}{h - \frac{\varepsilon}{1000}} , \tag{17}$$

"

Gat J R.: OXYGEN AND HYDROGEN ISOTOPES IN THE HYDROLOGIC CYCLE. Annu. rev. earth. planet. sci, 24, 225-262, doi:10.1146/annurev.earth.24.1.225, 1996.

Gibson, J. J., Birks, S. J., and Edwards, T.: Global prediction of da and d2h-d18o evaporation slopes for lakes and soil water accounting for seasonality. Global Biogeochem. Cy., 22, doi:10.1029/2007GB002997, 2008.

Hamilton, S. K., Bunn, S. E., Thoms, M. C., and Marshall, J. C.: Persistence of aquatic refugia between flow pulses in a dryland river system (Cooper Creek, Australia), Limnol. Oceanogr., 50, 743-754, doi:10.4319/lo.2005.50.3.0743, 2005.

Skrzypek, G., Mydłowski, A., Dogramaci, S., Hedley, P., Gibson, J. J., and Grierson, P. F.: Estimation of evaporative loss based on the stable isotope composition of water using Hydrocalculator, J. Hydrol., 523, 781-789, doi:10.1016/j.jhydrol.2015.02.010, 2015.

Sprenger, M., Tetzlaff, D., and Soulsby, C.: Soil water stable isotopes reveal evaporation dynamics at the soil–plant–atmosphere interface of the critical zone, Hydrol Earth Syst Sci, doi:10.5194/hess-21-3839-2017, 2017.

(5) However, the manuscript should be corrected by a native speaker (particularly the first part until Discussion).

Response: Thanks for your suggestion. Our paper is professionally edited by the Elsevier Language Editing Services.

(6) Finally, the title should be adapted since evaporating water is per se different from bulk soil water, and as such, the title does not reflect the process you seek to investigate.

Response: We modified our title to "Evaporating water is different from bulk soil water in $\delta^2$H and $\delta^{18}$O and implication for evaporation calculation"

**Specific comments**

L26 Please make clear that this is not a general statement but specific to the conditions of your experiment.

Response: Done. The small isotopic difference was added on P1 L29-32: "The soil evaporative water losses derived from EW isotopes were compared with those from BW. With a small isotopic difference between EW and BW, the evaporative water losses in the soil did not differ significantly ($p > 0.05$)."

L28 Which important implications?

Response: We modified the sentence on P2-2 L32-35: "Our results have important implications for quantifying evaporation processes using water stable isotopes. Future studies are needed to investigate how soil water isotopes partition differently between pores in soils with different pore size distributions and how this might affect soil evaporation estimation."

L41 "occupied" seems not the right term in this context.

Response: We replaced "occupied" with "filled". The detailed information was on P2 L51-53: "When large soil pores are filled with water, water in small pores does not participate in evaporation (Or and Lehmann, 2019; Zhang et al., 2015)."

Or, D. and Lehmann, P.: Surface evaporative capacitance: How soil type and rainfall characteristics affect global-scale surface evaporation, Water Resour. Res., 55, 519-539, doi:10.1029/2018WR024050, 2019.

Zhang, C., Li, L., and Lockington, D.: A physically based surface resistance model for evaporation from bare soils, Water Resour. Res., 51, 1084-1111, doi:10.1002/2014wr015490, 2015.

L45 Please rephrase the sentence.

Response: We rephrased the sentence on P2 L55-56: "With the progressive reduction of water in the larger pores, the evaporation rate gradually decreases."

L54 Large pores instead of pore.

Response: Done.

L37-59 This section should be moved to the methods.

Response: Thanks very much for your suggestion. In order to have a good flow in the introduction section, we kept this section in the introduction section. Combining the evaporation process and infiltration process, we raised our scientific question: Does evaporating water have different isotopic composition from bulk soil water? Thanks.

L82-84 This should be moved to the method section.

Response: Thanks for the suggestion. Done.

L84 Rephrase: This study may help to ….

Response: Thanks. The sentence was rephrased on P4 L93-94: "This study may help improve our understanding of soil evaporation and ecohydrological processes."

L98 Add values or signature in the sentence.

Response: Done. The values were added in P4 L105-109: "On August 26, 2016, the field was irrigated with 30 mm water ($\delta^2H$ = 49.87 ± 2.7 ‰, $\delta^{18}O$ = -9.40 ± 0.05 ‰, $n$ = 5) which was a mixture of tap water ($\delta^2H$ = -61.11 ‰, $\delta^{18}O$ = -9.42 ‰) and deuterium-enriched water (the $^2H$ concentration was 99.96%, $\delta^2H$ = 1.60 × $10^{10}$ ‰; Cambridge Isotope Laboratories, Inc., Tewksbury, MA, USA)."

L107 "secondary" evaporation

Response: Done.

L129-130 It is not clear whether the authors refer in this sentence to their own findings (in this case I would move the sentence to the results) or if there refer to other studies (in this case they should be cited). Moreover, the structure of the sentence is not clear and should be corrected.

Response: we were referring other studies. I separated the citations of the next sentence to two parts. The detailed information was added on P6 L152-157 : "However, in terms of isotopic compositions, the extracted water is generally depleted in heavy isotopes relative to the reference water, and the extent of depletion is affected by soil clay content and water content due to incomplete soil water extraction (Orlowski et al., 2016; Orlowski et al., 2013).To extract all water from a soil sample, a higher extraction temperature (>200 °C) might be desirable, especially for soils with substantial clay particles such as in the present study (clay content of 0.24 g g$^{-1}$) (Gaj et al., 2017a; Gaj et al., 2017b; Orlowski et al., 2018)."

Orlowski, N., Breuer, L., and McDonnell, J. J.: Critical issues with cryogenic extraction of soil water for stable isotope analysis, Ecohydrology, 9, 1-5, doi:10.1002/eco.1722, 2016.

Orlowski, N., Frede, H. G., Brüggemann, N., and Breuer, L.: Validation and application of a cryogenic vacuum extraction system for soil and plant water extraction for isotope analysis, J. Sens. Sens. Syst., 2, 179-193, doi:10.5194/jsss-2-179-2013, 2013.

Gaj, M., Kaufhold, S., Koeniger, P., Beyer, M., Weiler, M., and Himmelsbach, T.: Mineral mediated isotope fractionation of soil water, Rapid Commun. Mass Spectrom., 31, 269-280, doi:10.1002/rcm.7787, 2017a.

Gaj, M., Kaufhold, S., and McDonnell, J. J.: Potential limitation of cryogenic vacuum extractions and spiked experiments, Rapid Commun. Mass Spectrom., 31, 821-823, doi: 10.1002/rcm.7850, 2017b.

Orlowski, N., Breuer, L., Angeli, N., Boeckx, P., Brumbt, C., Cook, C. S., ... and McDonnell, J. J.: Interlaboratory comparison of cryogenic water extraction systems for stable isotope analysis of soil water, Hydrol Earth Syst Sci, 22, 3619-3637, doi:10.5194/hess-22-3619-2018, 2018.

L131: When are higher temperature needed? In case of higher clay content. This is not clear from the sentence. Could you provide soil texture information?

Response: Done. The soil texture information was added on P6 L155-157: "To extract all water from a soil sample, a higher extraction temperature (>200 °C) might be desirable, especially for soils with substantial clay particles such as in the present study (clay content of 0.24 g g$^{-1}$) (Gaj et al., 2017a; Gaj et al., 2017b; Orlowski et al., 2018)."

Gaj, M., Kaufhold, S., Koeniger, P., Beyer, M., Weiler, M., and Himmelsbach, T.: Mineral mediated isotope fractionation of soil water, Rapid Commun. Mass Spectrom., 31, 269-280, doi:10.1002/rcm.7787, 2017a.

Gaj, M., Kaufhold, S., and McDonnell, J. J.: Potential limitation of cryogenic vacuum extractions and spiked experiments, Rapid Commun. Mass Spectrom., 31, 821-823, doi: 10.1002/rcm.7850, 2017b.

Orlowski, N., Breuer, L., Angeli, N., Boeckx, P., Brumbt, C., Cook, C. S., ... and McDonnell, J. J.: Interlaboratory comparison of cryogenic water extraction systems for stable isotope analysis of soil water, Hydrol Earth Syst Sci, 22, 3619-3637, doi:10.5194/hess-22-3619-2018, 2018.

L147: "sub samples"

Response: Done.

L159 Use the present tense for referring to Tables and Figures.

Response: Done.

L167- : Why did you change the soil of the lysimeters. The reason is not apparent.

Response: The reason to change the soil of the lysimeters was added on P8 L203-207: "When evaporation occurs, unlike with soil outside the lysimeter, the soil within lysimeters is not replenished with water from deeper layers; thus, relative to soil outside the lysimeter, the soil water content within the lysimeters is generally smaller following continuous evaporation. Therefore, to represent the field soil conditions, the soil within the lysimeter was replaced every 4 days."

L222-228: Here, the introduction of the variables and equations is mixed up and difficult to follow. Please introduce each equation with its variables from top to bottom since this is an important aspect of your study.

Response: Thanks. We rewrote equations and the detailed information can be found on P10 L258-287:

[revised manuscript text omitted]

where $n$ is an intermediate variable and can be expressed as following,

$$n = \frac{-1.99 \alpha_1^*}{h - \frac{\varepsilon}{1000}} \ , \tag{18}$$

,,

L242. The article is missing: A general linear …

Response: Done.

L270: Delete "was".

Response: Done.

L277: What is meant by newly added water? The irrigation water? Please use the same terminology as before.

Response: Thanks. We used event water to describe precipitation event water and irrigation event water. We did revisions throughout the manuscript.

L300-306: style: delete some "therefore"

Response: Done.

L414: Could you explain how you estimated the value of 3.52 to result in significant differences.

Response: The method was added on P22 L481-484 : "The difference in Period II was 1.99 ‰ for $\delta^{18}$O. Nevertheless, the $\delta^{18}$O difference between EW and BW was too small to make a difference in the calculated evaporative water loss. However, hypothetically increasing the difference from 1.99 ‰ to 3.40 ‰, resulted in a significant difference in the calculated evaporative water loss ($p < 0.05$)."

L418: Do you mean matric potential?

Response: Yes. We changed it to matric potential on P23 L503-504: "Larger-pore water, preferred by evaporation, also has a relatively higher matric potential and flows more rapidly, and may thus be preferred by roots and dominate groundwater recharge (Sprenger et al., 2018)."

Sprenger, M., Tetzlaff, D., Buttle, J., Laudon, H., and Soulsby, C.: Water ages in the critical zone of long-term experimental sites in northern latitudes, Hydrol Earth Syst Sci, doi:10.5194/hess-22-3965-2018, 2018.

L436: Please make clear that this statement refers only to small differences in isotopic signals.

Response: Done. The information was added on P24 L522-524: "Our results indicate that although the isotopic composition in BW was significantly different from that in EW, the difference was too small to affect evaporative water loss calculation. However, a larger isotopic difference between the event and pre-event water may do."

**Comments from Reviewer: 2**

**Main comments:** Wang et al. sought to determine the contribution of bulk water from cryogenic extraction to evaporation water using stable isotopes of water. The team used a clever and practical method to collect evaporated water in a corn field and compared this to extracted bulk water throughout the growing season. Additionally, the authors applied a deuterium labeled irrigation to improve endmember resolution. Following the label, the evaporation and bulk water appears to decrease in 2H through time in similar overall values, whereas the 18O signature increases through time with significant differences between these two sampling domains. The authors interpret this to mean that, in this system, evaporation shows a strong preference for new water residing in large pores and that the source of evaporation differs from that of cryogenically extracted bulk water.

I think that both the aim and the results of this study are relevant and interesting. These kinds of experiments are severely lacking in modern hydrological sciences, and are needed to force the field to think openly about flow and mixing assumptions. However, there are numerous instances where the presentation and interpretation of the results make it difficult to judge the merit of the experiment, overall. I detail these discrepancies below. I think most of the necessary analyses have been conducted but I find it hard to accept without a substantial change to the current presentation and interpretations.

Response: Thank you. We will do our best to improve the quality of our manuscript.

**Specific Comments**

**Introduction to Evaporation Dynamics**

Lines 40-51: This section is a bit unclear. How exactly are the initial evaporation phases preferentially expressing larger pores? Yes, the larger pores connecting the deeper (more positive pore water pressure) source water to the near-surface may require higher contribution from higher conductivity ("larger") pores to sustain evaporation. However, it is unclear if the source of water vapor at the evaporation front is distinctly associated with larger pores, as smaller pores are dominated by stronger capillary forces (capillary > gravity + viscous forces) that maintain the gradient that links surface evaporation to deeper layers.

I think that this section needs to be made clearer which appears to be a critical point of the manuscript. I suggest providing a more detailed link to the literature, especially as these references (e.g, Ohr and Lehman + Zhang et al) do not make such obvious pore-scale distinctions.

Response: Yes, you are right. In the stage II, surface smaller pores (what we called is medium pores in our manuscript) link the surface evaporation to deeper soil layers, as the large pores are invaded by air. However, in stage I, large pore water dominates the evaporation flux; in stage III, surface small pore water (defined by the residual water in soil characteristic curve) and deeper larger pore water contribute to evaporation. Moreover, as pointed out by Zhang et al. (2015) "film water cannot be easily removed unless the local capillary water is dried out and the atmospheric demand for evaporation is strong. When the maximum volume of film water determines the residual water content." And the residual water is also used as the evaporation-insulated water in Or and Lehmann (2019). In order to be clearer, we modified our presentation for evaporation processes on P2 L46-62: "Water loss from soil progresses with air invasion into the soil in the order of large to small pores (Aminzadeh and Or, 2014; Lehmann and Or, 2009; Or et al., 2013). Soil pores can be divided into large, medium, and small pores. There is a minimum amount of small pore water at which liquid water in soil is still continuous or connected, below which liquid water is no longer connected, and vapor transport is the only way to further reduce water in soil. This water content is called the residual water content in the soil characteristic curve (Van Genuchten, 1980; Zhang et al., 2015). When large soil pores are filled with water, water in small pores does not participate in evaporation (Or and Lehmann, 2019; Zhang et al., 2015). Therefore, soil evaporation can be divided into three stages (Hillel, 1998; Or et al, 2013). Stage I: the evaporation front is in the surface soil, and water in large and medium pores participates in evaporation, but larger pores are the primary contributors. With the progressive reduction of water in the larger pores, the evaporation rate gradually decreases. Stage II: evaporation front is still in the surface soil, but larger pores are filled with air, water residing in the medium soil pores in the surface soil evaporates, and deep larger soil pores recharge the surface medium pores by capillary pull (Or and Lehmann, 2019), and the evaporation rate remains constant. Stage III: the hydraulic connectivity between the surface medium pores and deep large pores breaks, such that the evaporation front recedes into the subsurface soil. Water in the surface small pores and water in medium pores on the evaporation front evaporates. The evaporation rate decreases to a low value (Or et al, 2013)."

Aminzadeh, M. and Or, D.: Energy partitioning dynamics of drying terrestrial surfaces, J. Hydrol., 519, 1257-1270, doi:10.1016/j.jhydrol.2014.08.037, 2014.

Lehmann, P. and Or, D.: Evaporation and capillary coupling across vertical textural contrasts in porous media, Phys. Rev. E, 80, 046318, doi:10.1103/PhysRevE.80.046318, 2009.

Or, D., Lehmann, P., Shahraeeni, E., and Shokri, N.: Advances in soil evaporation physics—A review, Vadose Zone J, 12, 1-16, doi:10.2136/vzj2012.0163, 2013.

Van Genuchten, M. T.: A closed-form equation for predicting the hydraulic conductivity of unsaturated soils, Soil Sci Soc Am J, 44, 892-898, doi:10.2136/sssaj1980.03615995004400050002x, 1980.

Zhang, C., Li, L., and Lockington, D.: A physically based surface resistance model for evaporation from bare soils, Water Resour. Res., 51, 1084-1111, doi:10.1002/2014wr015490, 2015.

Or, D. and Lehmann, P.: Surface evaporative capacitance: How soil type and rainfall characteristics affect global-scale surface evaporation, Water Resour. Res., 55, 519-539, doi:10.1029/2018WR024050, 2019.

Hillel, D.: Environmental soil physics: Fundamentals, applications, and environmental considerations, Elsevier, 1998.

**Figures and Presentation**

Generally, it is difficult for the reader to interpret results from most of these figures. The labels of the figures are sporadic with non-intuitive descriptions in figure captions. Having to flip back and forth between plots and timelines to attribute dates with important time periods does not help (maybe get rid of dates, use time, and intuitive descriptors for each key time period?). Overall the quality of figures is often lacking. The exception is figure 8 which is well done. Please see my specific comments below (and attached file).

Response: Thank you for pointing out our issues on the figures. To be consistent, we changed the date to time i.e. days after precipitation/irrigation. But, for background information (Figure 4), we will keep using date. In order to be clearer, a larger font size was used for the axis labels of all the figures. Further, we also modified the descriptions in figure captions.

Also regarding the fractional evaporation:

Line 325: This gets a bit confusing.

1) how are you expressing the fraction of evaporated water source from both pools if equation 10 requires input from bulk water (i.e., this should work for just BW)?

Response: The calculation of evaporative water loss is based on isotopic mass balance of bulk soil water: The change of bulk soil water isotopic composition multiplied by the soil water storage reduction equals evaporation vapor isotopic composition multiplied by evaporation amount. Through Craig-Gordon model, the evaporation vapor isotopic composition can be obtained from the bulk soil water isotopic composition, or evaporating liquid water isotopic composition. Here we used bulk soil water isotopic

composition to express evaporating liquid water isotopic composition, and then used the latter to calculate evaporative water loss. Therefore, the evaporative water loss is expressed in equations 17 and 18 on P12 L293-297: "To calculate evaporative water loss from EW $\delta^{18}$O, we used BW to express EW and obtained the following formulas (Eqs. 18–19) for evaporative water loss.

$$f = 1 - \left[\frac{\delta_{BW} - \delta^* + n}{\delta_I - \delta^* + n}\right]^{\frac{1}{m}} , \tag{18}$$

where $n$ is an intermediate variable and can be expressed as follows:

$$n = \frac{-1.99\alpha_1^*}{h - \frac{\varepsilon}{1000}} , \tag{19}$$

"

2) why are you only comparing EW vs BW for $^{18}$O in period 2 and not $^2$H (or period 1)?

Response: In Period II, $\delta^2$H in BW and EW decreased with evaporation, meaning evaporation led to more "lighter" water in liquid phase, which is against our understanding of evaporation on soil water isotopes. Therefore, we cannot obtain evaporative water loss based on $\delta^2$H. The explanation was added on P19 L405-407: "However, the evaporative water loss could not be calculated from $\delta^2$H in BW or EW, as $\delta^2$H decreased as evaporation progressed (Fig. 5), which was inconsistent with the evaporation theory that soil evaporation enriches heavier water isotopes in the residual soil water." As mentioned in the previous comment, the isotopic relationship between EW and BW is needed in order to obtain EW isotopic composition from BW isotopic composition for evaporative water loss calculation. However, for evaporating water in Period I, we only have 4 data points, which do not allow for reliable regression. Therefore, we could not calculate evaporative water loss based on EW isotopic composition in Period I. The explanation was added on P19 L407-410: "Moreover, we could not calculate the evaporative water loss based on the isotopic composition of EW in Period I, as a reliable linear isotopic relationship between EW and BW could not be obtained from the four data points we had during the period."

3) Why make all of these sporadic comparisons and list one panel as not available. These points really detract from the meaning meant to be conveyed here.

Response: As mentioned above, the decrease of $\delta^2$H with increasing evaporation time against the evaporative theory, so we could not calculate evaporative water loss based on $\delta^2$H. The explanation was added on P19 L405-407: "However, the evaporative water loss could not be calculated from $\delta^2$H in BW

or EW, as $\delta^2$H decreased as evaporation progressed (Fig. 5), which was inconsistent with the evaporation theory that soil evaporation enriches heavier water isotopes in the residual soil water."

**Interpretation and Explanations**

Here are some key points:

Line 361: This is quite puzzling. How could you expect a difference in detected source in 18O between evaporation period and bulk water, when there is such a stronger end member separation in 2H? ~ 80 delta 2H per mil divided by instrument precision 0.2 = 400 units of detection versus almost no separation for 18O. If this finding is indeed true, I think it is worth discussing how you would see this in one isotopic signature (2H) and not 18O. Is it possible that the instrument precision of 2H was greatly reduced after the label (e.g., drift and memory effects) whereas we see a more correct version of 18O during phase 2? Would you have any data to calculate the precision of the analysis throughout the study period to confirm?

Response: Thanks for pointing this out. We analyzed the isotopic composition of condensation water, which was used to obtain the isotopic composition of evaporating water, as soon as we can after collecting it. We did the cryogenic extraction for bulk soil water including 0-5 cm soil and deep soil samples and subsequently analyzed the isotopic composition of bulk soil water. Therefore, our isotopic analysis started on 24-July-2016 and finished on 13-Jan.-2017. Three standard liquids LGR3C, LGR4C, and LGR5C were sequentially used to do the calibration, and three samples were analyzed following each standard. We did frequent analysis of standards to get rid of the instrument drift effect. In order to minimize the memory effect, every liquid was injected 6 times and the first 2 injections were discarded and the remaining 4 injections were averaged to obtain the isotopic value. Furthermore, the average $\delta^2$H and $\delta^{18}$O of LGR3C, LGR4C, and LGR5C throughout our study period were -97.34±0.020 ‰, -51.51±0.045 ‰, -9.26±0.025 ‰ and -13.42±0.003 ‰, -7.88±0.006 ‰, -2.72±0.003 ‰ (Mean±SE), respectively. The small standard error shows the excellent precision of our instrument throughout our study period. The detailed information was added on P8 L210-213: "Three liquid standards (LGR3C, LGR4C, and LGR5C and their respective $\delta^2$H = -97.30, -51.60, -9.20 ‰; $\delta^{18}$O = -13.39, -7.94, -2.69 ‰) were used sequentially for each of the three samples to remove the drift effect. To eliminate the memory effect, each sample was analyzed using six injections, of which only the last four injections were used to calculate the average value."

However, we did find that some of our isotopic values fall outside of those standards. The detailed information for extrapolation beyond the range of standards was added on P8 L213-222: "To check the effect of extrapolation beyond the range of standards, we performed a comparative experiment. In the experiment, 10 liquid samples with $\delta^2H$ varying from 0.14 to 107 ‰ and $\delta^{18}O$ from -1.75 to 12.24 ‰ were analyzed using LGR 3C, LGR 4C, and LGR 5C as standards (same with our former analysis) and were also analyzed using LGR 5C, GBW 04401 ($\delta^2H$ = -0.4 ‰, $\delta^{18}O$ = 0.32 ‰), and LGR E1 ($\delta^2H$ = 107 ‰, $\delta^{18}O$ = 12.24 ‰) as standards. The differences between the two sets of measurements were regressed with the sample isotope values obtained using LGR 5C, GBW 04401, and LGR E1 as standards, with a linear relationship of $\Delta^2H = -0.019\delta^2H-0.271$ (with $R^2$=1) and $\Delta^{18}O = -0.053\delta^{18}O-0.091$ (with $R^2$=1). We then applied the relationship and corrected the isotopic data that had $\delta^2H$ larger than -9.26 ‰ and $\delta^{18}O$ larger than -2.72 ‰. All the analyses in this study were based on the reanalyzed data."

Lines 373-375: Here is where the soil physics perspective matters. As you mention in your introduction (Lines 53-54) when tighter pores are filled with water (e.g., field capacity or wetter) the likelihood of preferential flow increases, as high porewater pressures force more water into large pores. However, under dry conditions (e.g., your irrigation event on 8/22) infiltrating water will initially fill these small pores, due to high matrix flux potential or a strong potential gradient between wetting front and dry soil. As the infiltration event proceeds, hydraulic length increases (e.g., depth of wetting front) driving down the infiltration rate (low gradient), the pore water pressures increase such that the air-entry pressure of large pores is exceeded, and then macropore or preferential flow ensues. Under the later phase gravitational forces exceed capillary "pull" into the matrix, increasing the likelihood of dual domain flow and separation between small and large pores.

The main point here is that dry conditions would likely facilitate preferential wetting of smaller pores due to strong capillary forces during initial infiltration. Thus, dry conditions could result in greater continuity between small and large pores. Having said this, preferential flow is known to happen under dry conditions too (especially in cracks) yet these conditions could really reduce the separation between the two pore domains. Note also that your introduction covers this process of preferential filling of small pores under dry conditions on Lines 52-53.

Please consider this point in your interpretation.

Response: Thank you for pointing out the controversial statement. We modified our interpretation on P3 L63-70: "Furthermore, pre-event soil water fills the smallest pores that are empty. When the event water amount is small, the empty small soil pores are filled with event water first (Beven and Germann, 1982; Brooks et al., 2010). However, when small pores are filled with water or when the amount of event water is large, the infiltration water preferentially enters larger pores and bypasses the saturated small pores (Beven and Germann, 1982; Booltink and Bouma, 1991; Sprenger and Allen, 2020). As larger pores have greater hydraulic conductivity, water residing in larger pores flows faster and drains first. Conversely, water residing in small pores drains lastly (Gerke and Van Genuchten, 1993; Phillips, 2010; Van Genuchten, 1980). Therefore, water in smaller pores has a longer residence time in the soil (Sprenger et al., 2019b)." and P21 L446-455: "For large precipitation events, event water preferentially infiltrates into the empty large pores because of their high hydraulic conductivity. The infiltrated water may partially or fully transfer to the surrounding empty smaller pores, thus bypassing the small soil pores that are filled with pre-event water at the point of water entry and along the infiltration pathway (Beven and Germann, 1982; Booltink and Bouma, 1991; Šimůnek and van Genuchten, 2008; Weiler and Naef, 2003; Zhang et al., 2019). In our experiment, the precipitation event on July 24, 2016, was 31 mm, and the irrigation event on August 26, 2016, was 30 mm, and both were large events. Because small pores were prefilled with pre-event water, we assumed that the new water filled large pores, and medium pores were likely filled by a mixture of pre-event and event water. Therefore, water in large pores was similar to the event water and water in the small pores was close to the pre-event water, i.e., old event water (Brooks et al., 2010; Sprenger et al., 2019a)."

Beven, K. and Germann, P.: Macropores and water flow in soils, Water Resour. Res., 18, 1311-1325, doi:10.1029/WR018i005p01311, 1982.

Brooks, J. R., Barnard, H. R., Coulombe, R., and McDonnell, J. J.: Ecohydrologic separation of water between trees and streams in a Mediterranean climate, Nat. Geosci., 3, 100-104, doi:10.1038/NGEO722, 2010.

Booltink, H. W. G. and Bouma, J.: Physical and morphological characterization of bypass flow in a well-structured clay soil, Soil Sci Soc Am J, 55, 1249-1254, doi:10.2136/sssaj1991.03615995005500050009x, 1991.

Sprenger, M. and Allen, S. T.: What ecohydrologic separation is and where we can go with it, Water Resour. Res., 56, e2020WR027238, doi:10.1029/2020wr027238, 2020.

Gerke, H. H. and Van Genuchten, M. T.: A dual-porosity model for simulating the preferential movement of water and solutes in structured porous media, Water Resour. Res., 29, 305-319, doi:10.1029/92WR02339, 1993.

Phillips, F. M.: Soil-water bypass, Nat. Geosci., 3, 77-78, doi:10.1038/ngeo762, 2010.

Van Genuchten, M. T.: A closed-form equation for predicting the hydraulic conductivity of unsaturated soils, Soil Sci Soc Am J, 44, 892-898, doi:10.2136/sssaj1980.03615995004400050002x, 1980.

Sprenger, M., Stumpp, C., Weiler, M., Aeschbach, W., Allen, S. T., Benettin, P., ... and McDonnell, J. J.: The demographics of water: A review of water ages in the critical zone, Rev. Geophys., 57, 800-834, doi:10.1029/2018rg000633, 2019b.

Weiler, M. and Naef, F.: An experimental tracer study of the role of macropores in infiltration in grassland soils, Hydrol Process, 17, 477-493, doi:10.1002/hyp.1136, 2003.

Sprenger, M., Llorens, P., Cayuela, C., Gallart, F., and Latron, J.: Mechanisms of consistently disconnected soil water pools over (pore) space and time, Hydrol Earth Syst Sci, 23, 1-18, doi:10.5194/hess-2019-143, 2019a.

Šimůnek, J. and van Genuchten, M.T.: Modeling Nonequilibrium Flow and Transport Processes Using HYDRUS. Vadose Zone J, 7, 782-797, doi:10.2136/vzj2007.0074, 2008.

Zhang, Z., Si, B., Li, H., Li, M.: Quantify piston and preferential water flow in deep soil using cl and soil water profiles in deforested apple orchards on the loess plateau, china. Water, 11, 2183, 2019.

Lines 381-382: Again, why exactly do you assume the small pores to only express old water? The average water content before irrigation was quite low (~ 0.15 in the upper 10 cm).

These 25 mm of irrigation could have filled ~7-10 cm of upper soil assuming a uniform wetting front and a conservative porosity of 0.45. Thus, the signature of infiltrating water alone could have muted the pre-event evaporation water source by >70%.

Response: You are correct. Event water could enter small pores too when they are empty at the time of water infiltration. We modified our presentations on P3 L63-70: "Furthermore, pre-event soil water fills the smallest pores that are empty. When the event water amount is small, the empty small soil pores are filled with event water first (Beven and Germann, 1982; Brooks et al., 2010). However, when small pores are filled with water or when the amount of event water is large, the infiltration water preferentially enters larger pores and bypasses the saturated small pores (Beven and Germann, 1982; Booltink and Bouma, 1991; Sprenger and Allen, 2020). As larger pores have greater hydraulic conductivity, water residing in larger pores flows faster and drains first. Conversely, water residing in small pores drains lastly (Gerke and Van Genuchten, 1993; Phillips, 2010; Van Genuchten, 1980). Therefore, water in smaller pores has a longer residence time in the soil (Sprenger et al., 2019b)."

[revised manuscript text omitted]

Lines 420-421: This is not consistent with Brooks et al. Brooks et al suggested that transpiration water and bulk soil were similar and that smaller pores with high residence time supplied this Ecohydrological flux.

Response: To the best of our knowledge, Brooks et al. (2010) suggested that large pores water will recharge streams (groundwater) and plant roots adsorb larger soil pores water, both of which making the progressively smaller soil pores contain water. In order to be clear, we added the information on P23 L506-508: "This is consistent with the findings of Brooks et al. (2010), as water-filled pores became progressively smaller after large-pore water percolates into streams (groundwater) and/or is adsorbed by plant roots, and can have broad ecohydrological implications."

Brooks, J. R., Barnard, H. R., Coulombe, R., and McDonnell, J. J.: Ecohydrologic separation of water between trees and streams in a Mediterranean climate, Nat. Geosci., 3, 100-104, doi:10.1038/NGEO722, 2010.

Specific comments:

Line 10: This reads like you are referring to the pool of water as being larger. "soil water from larger pores" is more clear and direct.

Response: Done.

Line 16: maybe distinguish this as "natural precipitation.." to be clear

Response: Done

Line 26: "…evaporation losses from .." from what?

Response: In order to make the meaning clear, we modified the sentence on P1 L29-32: "The soil evaporative water losses derived from EW isotopes were compared with those from BW. With a small isotopic difference between EW and BW, the evaporative water losses in the soil did not differ significantly ($p > 0.05$)."

Line: 27: "implicationS" (plural)

Response: Done.

Line 28: "process" Remove or make plural.

Response: Done.

Line 36: I do not think that these two previous sentences could be considered a full paragraph.

Response: We rephrased our presentation on P2 L37-45: "Terrestrial ecosystems receive water from precipitation and subsequently release all or part of the water to the atmosphere through evapotranspiration. The evapotranspiration process consumes approximately 25% of the incoming solar energy (Trenberth et al., 2009) and can be divided into two components: transpiration from plant leaves and evaporation from the soil surface. Soil evaporation varies from 10 to 60% of the total precipitation (Good et al., 2015; Oki and Kanae, 2006). Precise estimation of soil evaporative water loss relative to precipitation is critical for improving our knowledge of water budgets, plant water use efficiency, global ecosystem productivity, allocation of increasingly scarce water resources, and calibrating hydrological and climatic models (Kool et al., 2014; Oki and Kanae, 2006; Or et al., 2013; Or and Lehmann, 2019; Wang et al., 2014)."

[revised manuscript text omitted]

Lines 39-40: This sentence does not make sense as written. Also, it is not clear what you are trying to convey. Maybe you mean "minimum?"

Response: We modified the sentence on P2 L48-51: "There is a minimum amount of small pore water at which liquid water in soil is still continuous or connected, below which liquid water is no longer connected, and vapor transport is the only way to further reduce water in soil. This water content is called the residual water content in the soil characteristic curve (Van Genuchten, 1980; Zhang et al., 2015)."

Van Genuchten, M. T.: A closed-form equation for predicting the hydraulic conductivity of unsaturated soils, Soil Sci Soc Am J, 44, 892-898, doi:10.2136/sssaj1980.03615995004400050002x, 1980.

Zhang, C., Li, L., and Lockington, D.: A physically based surface resistance model for evaporation from bare soils, Water Resour. Res., 51, 1084-1111, doi:10.1002/2014wr015490, 2015.

Line 41: See earlier comment. Rephrase to water in smaller pores (or something like this). Please revise this throughout the manuscript

Response: Done.

Line 45: Try to be clear with this term "depleted," as this is also a study of water isotopes (e.g., isotopic depletion). Maybe choose a different word (e. g., drained).

Response: Done. We modified the presentation on P2 L55-56: "With the progressive reduction of water in the larger pores, the evaporation rate gradually decreases."

Lines 46-47: "capillary pumping" is never used in Or and Lehman (2019). This point is also unclear. Please specify.

Response: Thanks. We used "capillary pull" instead.

Line 60: use "infiltration" not "invasion"

Response: Done.

Line 71: "partitionING"

Response: Done.

Line 74: Okay, I think that the authors have used this small versus large pores enough to warrant a more specific reference. I suggest giving a more specific example of small versus large pores, especially here where vacuum pressure matters.

Response: Thanks for your concern. Commonly, we assume the cryogenic vacuum distillation with low pressure i.e. 0.2 Pa can extract water from all soil pores (large to small), as we described on P3 L84-85: "The isotopic composition of bulk soil water, which is extracted by cryogenic vacuum distillation, containing all pore water". Moreover, we defined the water in small pores on P2 L48-51: "There is a minimum amount of small pore water at which liquid water in soil is still continuous or connected, below which liquid water is no longer connected, and vapor transport is the only way to further reduce water in soil. This water content is called the residual water content in the soil characteristic curve (Van Genuchten, 1980; Zhang et al., 2015)."

Van Genuchten, M. T.: A closed-form equation for predicting the hydraulic conductivity of unsaturated soils, Soil Sci Soc Am J, 44, 892-898, doi:10.2136/sssaj1980.03615995004400050002x, 1980.

Zhang, C., Li, L., and Lockington, D.: A physically based surface resistance model for evaporation from bare soils, Water Resour. Res., 51, 1084-1111, doi:10.1002/2014wr015490, 2015.

Lines 77-78: Good point.

Response: Thank you.

Lines 84: "improve our understanding" works better? Does not make sense as written.

Response: Done. We rephrased our description on P4 L93-94: "This study may help improve our understanding of soil evaporation and ecohydrological processes."

Lines 133-135: Are these equations provided anywhere? Is the manuscript available for review. This seems to be an important detail.

Response: Thanks for your interest. As the paper that contains the related data was accepted by Hydrological Processes. The calibration equations from the paper are given on P6 L157-162: "Therefore, the water isotopic compositions obtained from our distillation system were subsequently corrected by calibration equations:

$\delta^2$H(post corrected)=$\delta^2$H(measured)-21.085*WC(water content)+5.144*CC(clay content)+5.944 and $\delta^{18}$O(post corrected)=$\delta^{18}$O(measured)-2.095*WC+0.783*CC+0.502 . The equations were obtained through a spiking experiment with 205 °C-oven-dried soils."

Lines 156-158: What exactly was measured here and what was calculated? Please state explicitly here and in the Supplemental file.

Response: As mentioned on P7 L179-183: "Hourly air and 0–5-cm soil temperature under the newly covered plastic film from September 10, 2016, to September 28, 2016, were measured using an E-type thermocouple (Omega Engineering, Norwalk, CT, USA) controlled by a CR1000 datalogger (Campbell Scientific, Inc., Logan, UT, USA). The 0–5-cm field soil temperature was measured during the whole field season using an ibutton device (DS1921G; Maxim Integrated, San Jose, CA, USA) at a frequency of 1 h." So, the air and 0-5 cm soil temperature under newly covered plastic film before September 10, 2016 were calculated and others were measured. The detailed information was added on P7 L183-194: "The 0–5-cm soil temperature and air temperature under the plastic film are required to calculate the evaporation ratios, but these measurements were not available before September 10, 2016. To obtain these temperature values, a regression equation was established between the measured 0–5-cm soil temperature values under the newly covered plastic film and those without plastic film covering from September 10, 2016, to September 28, 2016. We then used the equation to estimate 0–5-cm soil temperature under the newly covered plastic film before September 10, 2016, based on the ibutton-measured temperature of the 0–5-cm soil without the plastic film covering in the same period.

Subsequently, another regression equation was obtained between air temperature and 0–5-cm soil temperature from September 10, 2016, to September 28, 2016, both of which were under the newly covered plastic film. Then the air temperature under the newly covered plastic film before September 10, 2016, was estimated from the estimated 0–5-cm soil temperature under the newly covered plastic film. The regression equations are presented in the Supplement File."

Lines 170-176: Looks like you have 2 paragraphs with 2 sentences and no transition? Please fix this.

Response: Done. For better flow, we moved this part to P6 L135-145: "To obtain bulk soil density, field capacity, and residual water content, three 70-cm deep pits were dug at the end of the growing season. Stainless rings with a volume of 100 cm$^3$ (DIK-1801; Daiki Rika Kogyo Co., Ltd, Saitama, Japan) were pushed into the face of each soil pit at depths of 10, 20, 40, and 60 cm to obtain the soil samples. The soil samples were then saturated with distilled water, weighed, and placed in a high-speed centrifuge (CR21GII; Hitachi, Tokyo, Japan) with a centrifugation rotation velocity equivalent to a soil suction of 1 kPa for 10 min. The soil samples were weighed again to obtain the gravimetric water content at the aforementioned suction. This was repeated for suctions of 5, 10, 30, 50, 70, 100, 300, 500, and 700 kPa for 17, 26, 42, 49, 53, 58, 73, 81, and 85 min, respectively, to obtain the soil characteristic curve. After centrifugation, the soil samples were oven-dried and weighed to obtain the bulk soil density, which was used to convert gravimetric water content to volumetric water content."

Line 175: Should use "instrument" not "machine."

Response: Done.

Lines 201-202: Is it also possible that the plastic film itself can fractionate condensed water molecules? This point might be worth clarifying/considering at this stage.

Response: In order to avoid the secondary evaporation from the plastic film, we used a piece of plastic film without hole to cover the soil surface and collected the dew in the early morning. The detailed information was presented on P5 L114-115: "Subsequently, a piece of plastic film without holes (approximately 0.2 m$^2$, 40 and 50 cm) was used to cover the soil surface, with an extra 5 cm on each side." and P5 L116-122: "To eliminate the secondary evaporation of the condensation water, we first allowed evaporation and condensation to equilibrate for 2 days under the plastic film. Then, in the early

morning (approximately 7 a.m.), we collected the condensation water adhered to the underside of the plastic film using an injection syringe (Fig. 1a). The collected water was immediately transferred into a 1-mL glass vial. Therefore, it is reasonable to assume that the condensation water was in constant equilibrium with the evaporating water in the soil, and the water isotopes of evaporating water in the soil could be obtained from condensation water on the plastic film."

Line 246: "mean values." of what exactly?

Response: The information was added on P12 L303-304: "Further, Student's t-test (Knezevic, 2008) was used to compare two corresponding mean values of three replicates."

Knezevic, A.: Overlapping confidence intervals and statistical significance, StatNews: Cornell University Statistical Consulting Unit, 73, 2008.

Figure 4, Lines 258-259: This is very confusing . It looks like there are 4 periods. I suggest shading these these two areas with different colors or something similar.

Response: Thanks for the suggestions. Done.

[Figure]

Line 260: So the pink circles indicate when you compared bulk water versus evaporation water? Please clarify. Also were there no similar comparisons in Period 2?

Response: The detailed information was added on P17 L377-380: "In Period I, we compared the mean $\delta^2H$ and $\delta^{18}O$ values of all measurements within the green circle (Fig. 4) for both EW and BW. The mean $\delta^2H$ and $\delta^{18}O$ values for EW were significantly lower than those for BW ($p < 0.05$). Unfortunately, there were only four data points for EW, so we could not obtain a reliable isotopic relationship between EW and BW." For Period II, we compared the variation of isotopic composition in EW and BW with evaporation time i.e. the slopes and intercepts. The detailed information was presented on P17 L356-376: "The evaporation line, defined as the change in water isotopes with evaporation time in EW, was remarkably similar to that for BW (Fig. 5). For example, in Period II, $\delta^2H$ in both EW and BW decreased as evaporation proceeded, and both lines had a slope significantly smaller than zero ($p < 0.05$; Fig. 5b). This is contrary to our understanding that evaporation enriches $^2H$ in EW and BW. Moreover, it seemed that EW had higher $^2H$ vales than BW, but the slope and intercept of the EW evaporation line did not differ from that of the BW evaporation line ($p > 0.05$; Fig. 5b).

In period II, $\delta^{18}O$ in both EW and BW increased with evaporation time (Fig. 5d), and the slopes and intercepts significantly differed from zero ($p < 0.05$), indicating that evaporation, as expected, significantly enriched $^{18}O$ in EW and BW. However, there were some differences between EW and BW; $\delta^{18}O$ was consistently more depleted in EW than in BW during this period. Further regression analyses of $\delta^{18}O$ vs. time relationships in EW and BW in Period II indicated that though $\delta^{18}O$ vs. time in EW had the same slope as that in BW ($p > 0.05$), it had significantly smaller intercept than BW ($p < 0.05$). Thus, the linear relationship in $\delta^{18}O$ between EW and BW was given as $\delta^{18}O(EW) = \delta^{18}O(BW)-1.99$ (Fig. 5d). As is well known, the evaporation line ($\delta^{18}O$ vs. time) reflects the evaporative demand and the source water isotopic signature. First, the slopes of the evaporation lines represent the evaporative demand of the atmosphere. Given that EW and BW are under the same evaporative demand, their evaporation lines should have identical slopes. Second, the intercept of the evaporation line represents the isotopic signature of the initial evaporation water source. Therefore, in Period II, the intercepts of an $\delta^{18}O$ value of -1.76 ‰ for BW and -3.75 ‰ for EW represent the initial water sources of BW and EW, respectively. In other words, the sources of water for BW and EW had different isotopic compositions during Period II."

Line 263: What is the porosity?

Response: The porosity was added on P15 L323-325: "Soil water content in 0–5 cm reached field capacity (0.30 $cm^3$ $cm^{-3}$) with a volumetric water content of $0.30 \pm 0.007$ $cm^3$ $cm^{-3}$ and a porosity of $0.50 \pm 0.05$ $cm^3$ $cm^{-3}$ right after the first large precipitation event (July 24, 2016) and then decreased with evaporation time (grey bars in Fig. 4a)."

Line 265: Water contents can "jump"? :). please revise.

Response: Thanks. We changed it to "increased".

Line 266: Note that "Figure 4c" is not so clearly distinguished in the Figure. Would it be possible to move the letters e.g., "a)," "b)" to the left-hand side and increase the font size? Also, please refer to these sections directly in the figure captions.

Response: Thanks for the suggestion. Done.

[Figure]

Figure 4: The amount of precipitation, irrigation, and 0–5-cm bulk soil water content (a), $\delta^2$H and $\delta^{18}$O of precipitation and irrigation (b), $\delta^2$H of 0–5-cm bulk soil water and evaporating water (c), $\delta^{18}$O of 0–5-cm bulk soil water and evaporating water (d) at different times of the experimental period. Black arrows in panel (a) indicate dates when deep soil sampling took place, and the corresponding days after precipitation (irrigation) are indicated above the arrows. The two evaporation periods, marked by colored shades, include Period I from July 25, 2016, to August 25, 2016 (green) and Period II from August 27, 2016, to September 19, 2016 (cyan). Within the green circle in Period I, the mean ± standard error values were $\delta^2$H =-46.80 ± 1.07 ‰ and $\delta^{18}$O -3.22 ± 0.31 ‰ for 0–5-cm bulk soil water, and $\delta^2$H =-57.55 ± 2.60 ‰ and $\delta^{18}$O = -5.35 ± 0.22 ‰ for evaporating water.

Line 270: remove "was"

Response: Done.

Line 277: "Therefore" ??

Response: We modified it on P16 L341-343: "In summary, the event water in Period I was more depleted in heavy isotopes than in pre-event BW ($p < 0.05$). In Period II, the event water had a lower $\delta^{18}$O but a higher $\delta^2$H than pre-event BW ($p < 0.05$)."

Line 278: "relatively" should be "relative"

Response: Done.

Line 282: "resulting in.." this sentence has been cut off.

Response: Thanks. We omitted the comma. The detailed information was added on P16 L344-347: "The increase in $\delta^2$H and $\delta^{18}$O in BW had a significant linear relationship with evaporation time ($p < 0.05$; Fig. 5), suggesting that evaporation favored the lighter water isotopes from BW, resulting in greater $\delta^2$H and $\delta^{18}$O in BW."

Line 290: BW $^{18}$O also increased? Looks like there is a missing section??

Response: Yes, it is consistent with the last paragraph that describes BW in Period I. in order to be clearer, we jointed the two paragraphs.

Line 292: still describing period 2? Specify

Response: Yes, you are right. We added "in Period II" at the end of this sentence.

Line 306: Can you clarify why the period 1 EW and BW values are not shown together here? It looks like they would indicate a different source water for EW (minus one outlier)

Response: We added the description on P17 L377-380: "In Period I, we compared the mean $\delta^2$H and $\delta^{18}$O values of all measurements within the green circle (Fig. 4) for both EW and BW. The mean $\delta^2$H and $\delta^{18}$O values for EW were significantly lower than those for BW ($p < 0.05$). Unfortunately, there were only four data points for EW, so we could not obtain a reliable isotopic relationship between EW and BW." And on P19 L407-410 : "Moreover, we could not calculate the evaporative water loss based on the isotopic composition of EW in Period I, as a reliable linear isotopic relationship between EW and BW could not be obtained from the four data points we had during the period."

Line 321: I would really suggest getting rid of the dates here and using some intuitive representation in time (e.g., before irrigation, after irrigation, early period 1 etc..) It is difficult for the reader to discern what the various times mean and their relevance is not mentioned in the Figure 6 caption.

Response: Thanks for the suggestions. Done.

[Figure]

Figure 6: Temporal variation of deep soil water content, $\delta^2$H, $\delta^{18}$O, and lc-excess during Period I (upper panel) and Period II (lower panel). The precipitation event occurred on July 24, 2016, and the irrigation took place on August 26, 2016.

Line 342: "preferentially evaporated" is more grammatical correct.

Response: Done.

Line 354: "...THE evaporation period.."

Response: Done.

Line 362: difference in what? Please also specify for clarity.

Response: Done. The information was added on P20 L437-439: "No significant $\delta^2$H differences were observed between EW and BW in Period II (Fig. 5). However, there was a significant $\delta^{18}$O difference between EW and BW in Period II, and both $\delta^2$H and $\delta^{18}$O in EW differed from the respective values in BW in Period I (Figs. 4, 5)."

Line 365: "partitionING"

Response: Done.

Line 372: "...in larger pores than in small.."

Response: Done.

Line 408: difference did not make a difference?

Response: We revised the description on P22 L476-477: "4.3 Why the different isotopic compositions in evaporating water and bulk soil water did not make a difference in estimated evaporative water loss?"

Please also see my specific comments in the attached pdf, if needed.

Response: Thanks.

---

## Referee Report (RR1)

Evaporating water is different from bulk

soil water in 2H and 18O

**Summary:**

The authors have done a solid job of revising the manuscript for clarity, especially in terms of presentation and methodology. The manuscript now reads much better and many of the key findings are now more clearly conveyed. I think that the manuscript is nearly ready for publication but there are few important (and hopefully quickly addressable) points to consider from the soil physics perspective, as I detail below. These should, however, these should only be minor revisions.

**Page 2, Line 49**: "is no longer connected" should be "is hydraulically disconnected"

**Page 3, Line 63:** "pre-event" don't you mean "event"

**Page 3, Lines 65-80** (section starting with "…when small pores…"):

This a bit vague and does not appear to be totally correct.

Small pores being filled with water (and/or) and event water being "large" are not necessarily prerequisites for preferential flow.

You could easily have near-positive or positive pore water pressures near the soil surface displacing/filling water in small pores (without requiring all of the water to be channeled into preferential flow paths). But, then you could also have water from the soil matrix (smaller pores) mechanically displacing water in preferential flow paths (larger pores). In the former scenario pre-event "small" pore water could feasibly stay in the matrix, whereas in the later, the matrix water would presumably enter into preferential flow paths, with the potential to contribute heavily to evaporation (based on your three stage descriptions). See these papers:

Sklash, M., Beven, K., Gilman, K. & Darling, W. J. H. P. Isotope studies of pipeflow at Plynlimon, 465 Wales, UK. 10, 921-944 (1996).

Levy, B. S. & Germann, P. F. J. J. o. c. h. Kinematic wave approximation to solute transport along 559 preferred flow paths in soils. 3, 263-276 (1988).

Klaus, J., Zehe, E., Elsner, M., Külls, C. & McDonnell, J. Macropore flow of old water revisited: 460 experimental insights from a tile-drained hillslope. Hydrology and Earth System Sciences 17, 103-461 118 (2013).

Similarly, event water being "large" does not necessarily bear much relevance to the flow partitioning. The amount, in combination with intensity, is much more relevant as you could have a large rain event that is slowly infiltrating (low intensity) and conducive to matrix flow.

Rainfall intensity is particularly important because it may mean the difference between detecting similar or distinctly different water in matrix versus preferential flow paths.

See the general point:

Kumar, A., Kanwar, R. S., and Hallberg, G. R.: Separating preferential

and matrix flows using subsurface tile flow data, J. Environ

Sci. Heal. A, 32, 1711–1729, 1997.

Also see how this idea is applied more recently using stable isotopes of water at hillslope:

Klaus, J., Zehe, E., Elsner, M., Külls, C. & McDonnell, J. Macropore flow of old water revisited: 460 experimental insights from a tile-drained hillslope. Hydrology and Earth System Sciences 17, 103-461 118 (2013).

…and at the soil column scale, which details that these distinct pore scale separations may require extreme rainfall events (e.g.,50 y storm) and well-developed soil structure:

Radolinski, J., Pangle, L., Klaus, J. & Stewart, R. D. J. H. P. Testing the 'two water worlds' hypothesis under variable preferential flow conditions. Hydrological Processes 35, e14252 (2021).

I think the authors just need to be a little more explicit in their soil physical description/discussion. The 3-stage evaporation description reads better and makes much more sense now, which is very helpful. However, it may really benefit the manuscript to tighten-up the discussion of boundary conditions and how that influences mixing and/or the source of evaporation in soils.

**Page 21, Line 446:** "see my previous comment about large versus intense events. Please revise through the mansuscript.

**Page 21, Line 448:** Yes. But again, these do not necessarily have to be empty pores as displacement will likely occur simultaneously. (see comments, references, and discussion points that I made above).

**Page 21, Line 452:** "pre-filled with pre-event water". You were just describing bypass flow when small pores were empty. Now they are filled?

Please clarify.

**Page 23, Lines, 504 and 506-508:** Okay. Again, this does not appear to be strictly correct.

You are mentioning on Line 504 that "larger pore water" is preferred by plants and dominates groundwater recharge.

Then you mention that this is consistent with Brook et al. who categorically distinguish two water worlds as one tightly bound pool of soil water that supplies transpiration and another mobile pool that recharges groundwater and enters streams.

Brook et al., write on page 103: "Our results indicate that for this seasonally dry watershed within the Cascade Mountains of Oregon, soil water is separated into two water worlds: mobile water, which eventually enters the stream, and tightly bound water used by plants."

Please revise.

---

## Author Response (AR2)

Dear Dr. Orlowski,

Thank you very much for handling our manuscript entitled "Evaporating water is different from bulk soil water in  $\delta^2$ H and  $\delta^{18}$ O and implication for evaporation calculation". Corrections have been made and the detailed response to each comment from the reviewer is provided below. We tracked all the corrections within the revised manuscript and noted the new line numbers in this document where the corrections can be found in the revised manuscript.

We are looking forward to receiving your positive news.

Sincerely,

Hongxiu Wang Jingjing Jin Buli Cui Bingcheng Si Xiaojun Ma Mingyi Wen

**Comments from Editor**

The reviewer suggested some more minor revisions, which I would encourage you to follow. I think you did a great job with your revisions.

Response: Thanks for your positive comments. We did some revisions based on the suggestions from the reviewer, and the detailed response to each comment is provided below.

**Comments from Reviewer:**

**Main comment:**

The authors have done a solid job of revising the manuscript for clarity, especially in terms of presentation and methodology. The manuscript now reads much better and many of the key findings are now more clearly conveyed. I think that the manuscript is nearly ready for publication but there are few important (and hopefully quickly addressable) points to consider from the soil physics perspective, as I detail below. These should, however, these should only be minor revisions.

Response: Thanks for your positive comments and helpful suggestions. We did our best to enhance the soil physics perspective, including the infiltration processes, the boundary conditions, and the bypass flow occurrence in our experimental site.

**Specific comments**

Page 2, Line 49: "is no longer connected" should be "is hydraulically disconnected" Response: Done. Thanks.

Page 3, Line 63: "pre-event" don't you mean "event"

Response: The sentence was modified as P3 L64-65: "Furthermore, water in small pores and large pores may differ in isotopic compositions. As is well-known, pre-event soil water occupies the smallest pores."

Page 3, Lines 65-80 (section starting with "...when small pores..."):

This a bit vague and does not appear to be totally correct.

Small pores being filled with water (and/or) and event water being "large" are not necessarily prerequisites for preferential flow.

You could easily have near-positive or positive pore water pressures near the soil surface displacing/filling water in small pores (without requiring all of the water to be channeled into preferential flow paths). But, then you could also have water from the soil matrix (smaller pores) mechanically displacing water in preferential flow paths (larger pores). In the former scenario pre-event "small" pore water could feasibly stay in the matrix, whereas in the later, the matrix water would presumably enter

into preferential flow paths, with the potential to contribute heavily to evaporation (based on your three stage descriptions). See these papers:

Sklash, M., Beven, K., Gilman, K. & Darling, W. J. H. P. Isotope studies of pipeflow at Plynlimon, 465 Wales, UK. 10, 921-944 (1996).

Levy, B. S. & Germann, P. F. J. J. o. c. h. Kinematic wave approximation to solute transport along 559 preferred flow paths in soils. 3, 263-276 (1988).

Klaus, J., Zehe, E., Elsner, M., Külls, C. & McDonnell, J. Macropore flow of old water revisited: 460 experimental insights from a tile-drained hillslope. Hydrology and Earth System Sciences 17, 103-461 118 (2013).

Similarly, event water being "large" does not necessarily bear much relevance to the flow partitioning. The amount, in combination with intensity, is much more relevant as you could have a large rain event that is slowly infiltrating (low intensity) and conducive to matrix flow.

Rainfall intensity is particularly important because it may mean the difference between detecting similar or distinctly different water in matrix versus preferential flow paths.

See the general point:

Kumar, A., Kanwar, R. S., and Hallberg, G. R.: Separating preferential

and matrix flows using subsurface tile flow data, J. Environ

Sci. Heal. A, 32, 1711–1729, 1997.

Also see how this idea is applied more recently using stable isotopes of water at hillslope:

Klaus, J., Zehe, E., Elsner, M., Külls, C. & McDonnell, J. Macropore flow of old water revisited: 460 experimental insights from a tile-drained hillslope. Hydrology and Earth System Sciences 17, 103-461 118 (2013).

...and at the soil column scale, which details that these distinct pore scale separations may require extreme rainfall events (e.g., 50 y storm) and well-developed soil structure:

Radolinski, J., Pangle, L., Klaus, J. & Stewart, R. D. J. H. P. Testing the 'two water worlds' hypothesis under variable preferential flow conditions. Hydrological Processes 35, e14252 (2021).

I think the authors just need to be a little more explicit in their soil physical description/discussion. The 3-stage evaporation description reads better and makes much more sense now, which is very helpful. However, it may really benefit the manuscript to tighten-up the discussion of boundary conditions and how that influences mixing and/or the source of evaporation in soils. Response: Thanks for your fruitful suggestions. The infiltration processes were modified on P3 L64-87: "
[revised manuscript text omitted]

Page 21, Line 446: "see my previous comment about large versus intense events. Please revise through the manuscript.

Response: Done. "Large and intense events" was used.

Page 21, Line 448: Yes. But again, these do not necessarily have to be empty pores as displacement will likely occur simultaneously. (see comments, references, and discussion points that I made above).

Response: The infiltration processes is discussed on P22 L456-470: "For large and intense precipitation events, event water preferentially infiltrates into the empty large pores because of their high hydraulic conductivity. The infiltrated water may partially or fully transfer to the surrounding empty smaller pores, thus bypassing the small soil pores that are filled with pre-event water at the point of water entry and along the infiltration pathway (Beven and Germann, 1982; Booltink and Bouma, 1991; Šimůnek and van Genuchten, 2008; Weiler and Naef, 2003; Zhang et al., 2019). The bypass flow occurs universally (Lin 2010) and has also been reported in our experimental site, the Chinese Loess Plateau (Xiang et al., 2018; Zhang et al., 2019). In our experiment, the precipitation event on July 24, 2016, was 31 mm with the intensity of 10.3 mm h-1, and the irrigation event on August 26, 2016, was 30 mm with the intensity of 30 mm h-1, and both were sufficient to initiate bypass flow (> 10 mm h-1; Beven and Germann 1982; Kumar et al., 1997). The pre-event soil water content was close to residual water content (Section 3.1), indicating that small pores were prefilled with pre-event water. Thus, it is reasonable to assume that the new water filled large pores, and medium pores were likely filled by a mixture of pre-event and event water. Therefore, water in large pores was similar to the event water and water in the small pores was close to the pre-event water, i.e., old event water (Brooks et al., 2010; Sprenger et al., 2019a)."

- Beven, K. and Germann, P.: Macropores and water flow in soils, Water Resour. Res., 18, 1311-1325, doi:10.1029/WR018i005p01311, 1982.
- Booltink, H. W. G. and Bouma, J.: Physical and morphological characterization of bypass flow in a wellstructured clay soil, Soil Sci Soc Am J, 55, 1249-1254, doi:10.2136/sssaj1991.03615995005500050009x, 1991.
- Šimůnek, J. and van Genuchten, M.T.: Modeling Nonequilibrium Flow and Transport Processes Using HYDRUS. Vadose Zone J, 7, 782-797, doi:10.2136/vzj2007.0074, 2008.
- Weiler, M. and Naef, F.: An experimental tracer study of the role of macropores in infiltration in grassland soils, Hydrol Process, 17, 477-493, doi:10.1002/hyp.1136, 2003.
- Zhang, Z., Si, B., Li, H., Li, M.: Quantify piston and preferential water flow in deep soil using cl and soil water profiles in deforested apple orchards on the loess plateau, china. Water, 11, 2183, 2019.
- Lin, H.: Linking principles of soil formation and flow regimes, J. Hydrol., 393(1-2), 3-19, doi:10.1016/j.jhydrol.2010.02.013, 2010.
- Xiang, W., Si, B. C., Biswas, A., and Li, Z.: Quantifying dual recharge mechanisms in deep unsaturated zone of chinese loess plateau using stable isotopes, Geoderma, 337, 773-781, doi:10.1016/j.geoderma.2018.10.006, 2018.
- Kumar, A., Kanwar, R. S., and Hallberg, G. R.: Separating preferential and matrix flows using subsurface tile flow data, J. Environ. Health Sci. Eng., Part A: Environmental Science & Engineering & Toxic & Hazardous Substance Control, doi:10.1080/10934529709376639, 1997
- Brooks, J. R., Barnard, H. R., Coulombe, R., and McDonnell, J. J.: Ecohydrologic separation of water between trees and streams in a Mediterranean climate, Nat. Geosci., 3, 100-104, doi:10.1038/NGEO722, 2010.
- Sprenger, M., Llorens, P., Cayuela, C., Gallart, F., and Latron, J.: Mechanisms of consistently disconnected soil water pools over (pore) space and time, Hydrol Earth Syst Sci, 23, 1-18, doi:10.5194/hess-2019-143, 2019a.

Page 21, Line 452: "pre-filled with pre-event water". You were just describing bypass flow when small pores were empty. Now they are filled?

Please clarify.

Response: Yes, pre-event water filled the small soil pores. The reason was added on P22 L465-470 : "The pre-event soil water content was close to residual water content (Section 3.1), indicating that small pores were prefilled with pre-event water. Thus, it is reasonable to assume that the new water filled large pores, and medium pores were likely filled by a mixture of pre-event and event water. Therefore, water in large pores was similar to the event water and water in the small pores was close to the pre-event water, i.e., old event water (Brooks et al., 2010; Sprenger et al., 2019a)."

Brooks, J. R., Barnard, H. R., Coulombe, R., and McDonnell, J. J.: Ecohydrologic separation of water

between trees and streams in a Mediterranean climate, Nat. Geosci., 3, 100-104, doi:10.1038/NGEO722, 2010.

Sprenger, M., Llorens, P., Cayuela, C., Gallart, F., and Latron, J.: Mechanisms of consistently disconnected soil water pools over (pore) space and time, Hydrol Earth Syst Sci, 23, 1-18, doi:10.5194/hess-2019-143, 2019a.

Page 23, Lines, 504 and 506-508: Okay. Again, this does not appear to be strictly correct.

You are mentioning on Line 504 that "larger pore water" is preferred by plants and dominates groundwater recharge.

Then you mention that this is consistent with Brook et al. who categorically distinguish two water worlds as one tightly bound pool of soil water that supplies transpiration and another mobile pool that recharges groundwater and enters streams.

Brook et al., write on page 103: "Our results indicate that for this seasonally dry watershed within the Cascade Mountains of Oregon, soil water is separated into two water worlds: mobile water, which eventually enters the stream, and tightly bound water used by plants."

Please revise.

Response: Thanks very much for your suggestion. We really appreciate it. We revised the text on P24 L521-527: "This is inconsistent with Brooks et al. (2010), who separated soil water into two water worlds: mobile water, which eventually enters the stream, and tightly bound water used by plants. In our study, soil water content was below field capacity and thus according to Brooks et al. (2010), all water in our soil is "tightly bound water", including the large pore water we discussed above. Therefore, in our study, the larger pore water is still under the field capacity, the water that percolates into streams (groundwater) rather slowly and/or is adsorbed by plant roots, which has broad ecohydrological implications."

Brooks, J. R., Barnard, H. R., Coulombe, R., and McDonnell, J. J.: Ecohydrologic separation of water between trees and streams in a Mediterranean climate, Nat. Geosci., 3, 100-104, doi:10.1038/NGEO722, 2010.